# Alternating Estimation for Structured High-Dimensional Multi-Response Models

**Sheng Chen**        **Arindam Banerjee**
Dept. of Computer Science & Engineering
University of Minnesota, Twin Cities
{shengc,banerjee}@cs.umn.edu

## Abstract

We consider the problem of learning high-dimensional multi-response linear models with structured parameters. By exploiting the noise correlations among different responses, we propose an alternating estimation (AltEst) procedure to estimate the model parameters based on the generalized Dantzig selector (GDS). Under suitable sample size and resampling assumptions, we show that the error of the estimates generated by AltEst, with high probability, converges linearly to certain minimum achievable level, which can be tersely expressed by a few geometric measures, such as Gaussian width of sets related to the parameter structure. To the best of our knowledge, this is the first non-asymptotic statistical guarantee for such AltEst-type algorithm applied to estimation with general structures.

## 1 Introduction

Multi-response (a.k.a. multivariate) linear models [2, 8, 20, 21] have found numerous applications in real-world problems, e.g. expression quantitative trait loci (eQTL) mapping in computational biology [28], land surface temperature prediction in climate informatics [17], neural semantic basis discovery in cognitive science [30], etc. Unlike simple linear model where each response is a scalar, one obtains a *response vector* at each observation in multi-response model, given as a (noisy) linear combinations of predictors, and the parameter (i.e., coefficient vector) to learn can be either response-specific (i.e., allowed to be different for every response), or shared by all responses. The multi-response model has been well studied under the context of the multi-task learning [10], where each response is coined as a *task*. In recent years, the multi-task learning literature have largely focused on exploring the parameter structure across tasks via convex formulations [15, 3, 26]. Another emphasis area in multi-response modeling is centered around the exploitation of the noise correlation among different responses [35, 36, 29, 40, 42], instead of assuming that the noise is independent for each response. To be specific, we consider the following multi-response linear models with $m$ real-valued outputs,

$$\mathbf{y}_i = \mathbf{X}_i \boldsymbol{\theta}^* + \boldsymbol{\eta}_i, \qquad \boldsymbol{\eta}_i \sim \mathcal{N}(\mathbf{0}, \boldsymbol{\Sigma}_*) , \tag{1}$$

where $\mathbf{y}_i \in \mathbb{R}^m$ is the response vector, $\mathbf{X}_i \in \mathbb{R}^{m \times p}$ consists of $m$ $p$-dimensional feature vectors, and $\boldsymbol{\eta}_i \in \mathbb{R}^m$ is a noise vector sampled from a multivariate zero-mean Gaussian distribution with covariance $\boldsymbol{\Sigma}_*$. For simplicity, we assume $\text{Diag}(\boldsymbol{\Sigma}_*) = \mathbf{I}_{m \times m}$ throughout the paper. The $m$ responses share the same underlying parameter $\boldsymbol{\theta}^* \in \mathbb{R}^p$, which corresponds to the so-called *pooled model* [19]. In fact, this seemingly restrictive setting is general enough to encompass the model with response-specific parameters, which can be realized by block-diagonalizing rows of $\mathbf{X}_i$ and stacking all coefficient vectors into a "long" vector. Under the assumption of correlated noise, the true noise covariance structure $\boldsymbol{\Sigma}_*$ is usually unknown. Therefore it is typically required to estimate the parameter $\boldsymbol{\theta}^*$ along with the covariance $\boldsymbol{\Sigma}_*$. In practice, we observe $n$ data points, denoted by

$\mathcal{D} = \{(\mathbf{X}_i, \mathbf{y}_i)\}_{i=1}^n$, and the maximum likelihood estimator (MLE) is simply as follows,

$$\left(\hat{\boldsymbol{\theta}}_{\text{MLE}}, \hat{\boldsymbol{\Sigma}}_{\text{MLE}}\right) = \underset{\boldsymbol{\theta} \in \mathbb{R}^p, \, \boldsymbol{\Sigma} \succeq 0}{\operatorname{argmin}} \frac{1}{2} \log |\boldsymbol{\Sigma}| + \frac{1}{2n} \sum_{i=1}^n \left\| \boldsymbol{\Sigma}^{-\frac{1}{2}} (\mathbf{y}_i - \mathbf{X}_i \boldsymbol{\theta}) \right\|_2^2 \tag{2}$$

Although being convex w.r.t. either $\boldsymbol{\theta}$ or $\boldsymbol{\Sigma}$ when the other is fixed, the optimization problem associated with the MLE is jointly *non-convex* for $\boldsymbol{\theta}$ and $\boldsymbol{\Sigma}$. A popular approach to dealing with such problem is *alternating minimization* (AltMin), i.e., alternately solving for $\boldsymbol{\theta}$ (and $\boldsymbol{\Sigma}$) while keeping $\boldsymbol{\Sigma}$ (and $\boldsymbol{\theta}$) fixed. The AltMin algorithm for (2) iteratively performs two simple steps, solving least squares for $\boldsymbol{\theta}$ and computing empirical noise covariance for $\boldsymbol{\Sigma}$. Recent work [24] has established the non-asymptotic error bound of this approach for (2) with a brief extension to sparse parameter setting using iterative hard thresholding method [25]. But they did not allow more general structure of the parameter. Previous works [35, 29, 33] also considered the regularized MLE approaches for multi-response models with sparse parameters, which are solved by AltMin-type algorithms as well. Unfortunately, *none* of those works provide *finite-sample* statistical guarantees for their algorithms. AltMin technique has also been applied to many other problems, such as matrix completion [23], sparse coding [1], and mixed linear regression [41], with provable performance guarantees. Despite the success of AltMin, most existing works are dedicated to recovering unstructured sparse or low-rank parameters, with little attention paid to general structures, e.g., overlapping sparsity [22], hierarchical sparsity [27], $k$-support sparsity [4], etc.

In this paper, we study the multi-response linear model in high-dimensional setting, i.e., sample size $n$ is smaller than the problem dimension $p$, and the coefficient vector $\boldsymbol{\theta}^*$ is assumed to possess a general low-complexity structure, which can be essentially captured by certain *norm* $\|\cdot\|$ [5]. Structured estimation using norm regularization/minimization has been extensively studied for simple linear models over the past decade, and recent advances manage to characterize the estimation error for convex approaches including Lasso-type (regularized) [38, 31, 6] and Dantzig-type (constrained) estimator [7, 12, 14], via a few simple *geometric measures*, e.g., *Gaussian width* [18, 11] and *restricted norm compatibility* [31, 12]. Here we propose an *alternating estimation* (AltEst) procedure for finding the true parameters, which essentially alternates between estimating $\boldsymbol{\theta}$ through the generalized Dantzig selector (GDS) [12] using norm $\|\cdot\|$ and computing the approximate empirical noise covariance for $\boldsymbol{\Sigma}$. Our analysis puts no restriction on what the norm can be, thus the AltEst framework is applicable to general structures. In contrast to AltMin, our AltEst procedure *cannot* be casted as a minimization of some joint objective function for $\boldsymbol{\theta}$ and $\boldsymbol{\Sigma}$, thus is conceptually more general than AltMin. For the proposed AltEst, we provide the statistical guarantees for the iterate $\hat{\boldsymbol{\theta}}_t$ with the *resampling* assumption (see Section 2), which may justify the applicability of AltEst technique to other problems without joint objectives for two set of parameters. Specifically, we show that with overwhelming probability, the estimation error $\|\hat{\boldsymbol{\theta}}_t - \boldsymbol{\theta}^*\|_2$ for generally structured $\boldsymbol{\theta}^*$ converges *linearly* to a *minimum achievable error* given sub-Gaussian design under moderate sample size. With a straightforward intuition, this minimum achievable error can be tersely expressed by the aforementioned geometric measures which simply depend on the structure of $\boldsymbol{\theta}^*$. Moreover, our analysis implies the error bound for single response high-dimensional models as a by-product [12]. Note that the analysis in [24] focuses on the expected prediction error $\mathbb{E}[\boldsymbol{\Sigma}_*^{-1/2} \mathbf{X}(\hat{\boldsymbol{\theta}}_t - \boldsymbol{\theta}^*)]$ for unstructured $\boldsymbol{\theta}^*$, which is related but different from our $\|\hat{\boldsymbol{\theta}}_t - \boldsymbol{\theta}^*\|_2$ for generally structured $\boldsymbol{\theta}^*$. Compared with the error bound derived for unstructured $\boldsymbol{\theta}^*$ in [24], our result also yields better dependency on sample size by removing the $\log n$ factor, which seems unnatural to appear.

The rest of the paper is organized as follows. We elaborate our AltEst algorithm in Section 2, along with the resampling assumption. In Section 3, we present the statistical guarantees for AltEst. We provide experimental results in Section 4 to support our theoretical development. Finally we conclude in Section 5. Due to space limitations, all proofs are deferred to the supplementary material.

## 2 Alternating Estimation for High-Dimensional Multi-Response Models

Given the high-dimensional setting for (1), it is natural to consider the regularized MLE for (1) by adding the norm $\|\cdot\|$ to (2), which captures the structural information of $\boldsymbol{\theta}^*$ in (1),

$$\left(\hat{\boldsymbol{\theta}}, \hat{\boldsymbol{\Sigma}}\right) = \underset{\boldsymbol{\theta} \in \mathbb{R}^p, \, \boldsymbol{\Sigma} \succeq 0}{\operatorname{argmin}} \frac{1}{2} \log |\boldsymbol{\Sigma}| + \frac{1}{2n} \sum_{i=1}^n \left\| \boldsymbol{\Sigma}^{-\frac{1}{2}} (\mathbf{y}_i - \mathbf{X}_i \boldsymbol{\theta}) \right\|_2^2 + \gamma_n \|\boldsymbol{\theta}\| , \tag{3}$$

where $\gamma_n$ is a tuning parameter. Using AltMin the update of (3) can be given as

$$\hat{\boldsymbol{\theta}}_t = \underset{\boldsymbol{\theta} \in \mathbb{R}^p}{\operatorname{argmin}} \ \frac{1}{2n} \sum_{i=1}^{n} \left\| \hat{\boldsymbol{\Sigma}}_{t-1}^{-\frac{1}{2}} (\mathbf{y}_i - \mathbf{X}_i \boldsymbol{\theta}) \right\|_2^2 + \gamma_n \|\boldsymbol{\theta}\| \tag{4}$$

$$\hat{\boldsymbol{\Sigma}}_t = \frac{1}{n} \sum_{i=1}^{n} \left( \mathbf{y}_i - \mathbf{X}_i \hat{\boldsymbol{\theta}}_t \right) \left( \mathbf{y}_i - \mathbf{X}_i \hat{\boldsymbol{\theta}}_t \right)^T \tag{5}$$

The update of $\hat{\boldsymbol{\theta}}_t$ is basically solving a regularized least squares problem, and the new $\hat{\boldsymbol{\Sigma}}_t$ is obtained by computing the approximated empirical covariance of the residues evaluated at $\hat{\boldsymbol{\theta}}_t$. In this work, we consider an alternative to (4), the generalized Dantzig selector (GDS) [12], which is given by

$$\hat{\boldsymbol{\theta}}_t = \underset{\boldsymbol{\theta} \in \mathbb{R}^p}{\operatorname{argmin}} \ \|\boldsymbol{\theta}\| \quad \text{s.t.} \quad \left\| \frac{1}{n} \sum_{i=1}^{n} \mathbf{X}_i^T \hat{\boldsymbol{\Sigma}}_{t-1}^{-1} (\mathbf{X}_i \boldsymbol{\theta} - \mathbf{y}_i) \right\|_* \leq \gamma_n \ , \tag{6}$$

where $\| \cdot \|_*$ is the *dual norm* of $\| \cdot \|$. Compared with (4), GDS has nicer geometrical properties, which is favored in the statistical analysis. More importantly, since iteratively solving (6) followed by covariance estimation (5) no longer minimizes a specific objective function jointly, the updates go beyond the scope of AltMin, leading to our broader alternating estimation (AltEst) framework, i.e., alternately estimating one parameter by suitable approaches while keeping the other fixed. For the ease of exposition, we focus on the $m \leq n$ scenario, so that $\hat{\boldsymbol{\Sigma}}_t$ can be easily computed in closed form as shown in (5). When $m > n$ and $\boldsymbol{\Sigma}_*^{-1}$ is sparse, it is beneficial to directly estimate $\boldsymbol{\Sigma}_*^{-1}$ using more advanced estimators [16, 9]. Especially the CLIME estimator [9] enjoys certain desirable properties, which fits into our AltEst framework but not AltMin, and our AltEst analysis *does not* rely on the particular estimator we use to estimate noise covariance or its inverse. The algorithmic details are given in Algorithm 1, for which it is worth noting that every iteration $t$ uses independent new samples, $\mathcal{D}_{2t-1}$ and $\mathcal{D}_{2t}$ in Step 3 and 4, respectively. This assumption is known as *resampling*, which facilitates the theoretical analysis by removing the statistical dependency between iterates. Several existing works benefit from such assumption when analyzing their AltMin-type algorithms [23, 32, 41]. Conceptually resampling can be implemented by partitioning the whole dataset into $T$ subsets, though it is unusual to do so in practice. Loosely speaking, AltEst (AltMin) with resampling is an approximation of the practical AltEst (AltMin) with a single dataset $\mathcal{D}$ used by all iterations. For AltMin, attempts have been made to directly analyze its practical version without resampling, by studying the properties of the joint objective [37], which come at the price of invoking highly sophisticated mathematical tools. This technique, however, might fail to work for AltEst since the procedure is not even associated with a joint objective. In the next section, we will leverage such resampling assumption to show that the error of $\hat{\boldsymbol{\theta}}_t$ generated by Algorithm 1 will converge to a small value with high probability. We again emphasize that the AltEst framework may work for other suitable estimators for $(\boldsymbol{\theta}^*, \boldsymbol{\Sigma}_*)$ although (5) and (6) are considered in our analysis.

---

**Algorithm 1** Alternating Estimation with Resampling

---

**Input:** Number of iterations $T$, Datasets $\mathcal{D}_1 = \{(\mathbf{X}_i, \mathbf{y}_i)\}_{i=1}^n, \ldots, \mathcal{D}_{2T} = \{(\mathbf{X}_i, \mathbf{y}_i)\}_{i=(2T-1)n+1}^{2Tn}$

1: Initialize $\hat{\boldsymbol{\Sigma}}_0 = \mathbf{I}_{m \times m}$
2: **for** $t := 1$ to $T$ **do**
3:      Solve the GDS (6) for $\hat{\boldsymbol{\theta}}_t$ using dataset $\mathcal{D}_{2t-1}$
4:      Compute $\hat{\boldsymbol{\Sigma}}_t$ according to (5) using dataset $\mathcal{D}_{2t}$
5: **end for**
6: **return** $\hat{\boldsymbol{\theta}}_T$

---

## 3 Statistical Guarantees for Alternating Estimation

In this section, we establish the statistical guarantees for our AltEst algorithm. The road map for the analysis is to first derive the error bounds separately for both (5) and (6), and then combine them through AltEst procedure to show the error bound of $\hat{\boldsymbol{\theta}}_t$. Throughout the analysis, the design $\mathbf{X}$ is assumed to centered, i.e., $\mathbb{E}[\mathbf{X}] = \mathbf{0}_{m \times p}$. $\lambda_{\max}(\cdot)$ and $\lambda_{\min}(\cdot)$ are used to denote the largest and smallest eigenvalue of a real symmetric matrix. Before presenting the results, we provide some basic but important concepts. First of all, we give the definition of sub-Gaussian matrix $\mathbf{X}$.

**Definition 1 (Sub-Gaussian Matrix)** $\mathbf{X} \in \mathbb{R}^{m \times p}$ is sub-Gaussian if the $\psi_2$-norm below is finite,

$$\|\mathbf{X}\|_{\psi_2} = \sup_{\mathbf{v} \in \mathbb{S}^{p-1},\ \mathbf{u} \in \mathbb{S}^{m-1}} \left\| \left\| \mathbf{v}^T \boldsymbol{\Gamma}_{\mathbf{u}}^{-\frac{1}{2}} \mathbf{X}^T \mathbf{u} \right\| \right\|_{\psi_2} \leq \kappa < +\infty \,, \tag{7}$$

where $\boldsymbol{\Gamma}_{\mathbf{u}} = \mathbb{E}[\mathbf{X}^T \mathbf{u} \mathbf{u}^T \mathbf{X}]$. Further we assume there exist constants $\mu_{\min}$ and $\mu_{\max}$ such that

$$0 < \mu_{\min} \leq \lambda_{\min}(\boldsymbol{\Gamma}_{\mathbf{u}}) \leq \lambda_{\max}(\boldsymbol{\Gamma}_{\mathbf{u}}) \leq \mu_{\max} < +\infty \,, \quad \forall\, \mathbf{u} \in \mathbb{S}^{m-1} \tag{8}$$

The definition (7) is also used in earlier work [24], which assumes the left end of (8) implicitly. Lemma 1 gives an example of sub-Gaussian $\mathbf{X}$, showing that condition (7) and (8) are reasonable.

**Lemma 1** *Assume that $\mathbf{X} \in \mathbb{R}^{m \times p}$ has dependent anisotropic rows such that $\mathbf{X} = \boldsymbol{\Xi}^{\frac{1}{2}} \tilde{\mathbf{X}} \boldsymbol{\Lambda}^{\frac{1}{2}}$, where $\boldsymbol{\Xi} \in \mathbb{R}^{m \times m}$ encodes the dependency between rows, $\tilde{\mathbf{X}} \in \mathbb{R}^{m \times p}$ has independent isotropic rows, and $\boldsymbol{\Lambda} \in \mathbb{R}^{p \times p}$ introduces the anisotropy. In this setting, if each row of $\tilde{\mathbf{X}}$ satisfies $\|\tilde{\mathbf{x}}_i\|_{\psi_2} \leq \tilde{\kappa}$, then condition (7) and (8) hold with $\kappa = C\tilde{\kappa}$, $\mu_{\min} = \lambda_{\min}(\boldsymbol{\Xi})\lambda_{\min}(\boldsymbol{\Lambda})$, and $\mu_{\max} = \lambda_{\max}(\boldsymbol{\Xi})\lambda_{\max}(\boldsymbol{\Lambda})$.*

The recovery guarantee of GDS relies on an important notion called *restricted eigenvalue* (RE). In multi-response setting, it is defined jointly for designs $\mathbf{X}_i$ and a noise covariance $\boldsymbol{\Sigma}$ as follows.

**Definition 2 (Restricted Eigenvalue Condition)** The designs $\mathbf{X}_1, \mathbf{X}_2, \ldots, \mathbf{X}_n$ and the covariance $\boldsymbol{\Sigma}$ together satisfy the restricted eigenvalue condition for set $\mathcal{A} \subseteq \mathbb{S}^{p-1}$ with parameter $\alpha > 0$, if

$$\inf_{\mathbf{v} \in \mathcal{A}} \mathbf{v}^T \left( \frac{1}{n} \sum_{i=1}^{n} \mathbf{X}_i^T \boldsymbol{\Sigma}^{-1} \mathbf{X}_i \right) \mathbf{v} \geq \alpha \,. \tag{9}$$

Apart from RE condition, the analysis of GDS is carried out on the premise that tuning parameter $\gamma_n$ is suitably selected, which we define as "admissible".

**Definition 3 (Admissible Tuning Parameter)** The $\gamma_n$ for GDS (6) is said to be *admissible* if $\gamma_n$ is chosen such that $\boldsymbol{\theta}^*$ belongs to the constraint set, i.e.,

$$\left\| \frac{1}{n} \sum_{i=1}^{n} \mathbf{X}_i^T \boldsymbol{\Sigma}^{-1} (\mathbf{X}_i \boldsymbol{\theta}^* - \mathbf{y}_i) \right\|_* = \left\| \frac{1}{n} \sum_{i=1}^{n} \mathbf{X}_i^T \boldsymbol{\Sigma}^{-1} \boldsymbol{\eta}_i \right\|_* \leq \gamma_n \tag{10}$$

For structured estimation, one also needs to characterize the structural complexity of $\boldsymbol{\theta}^*$, and an appropriate choice is the Gaussian width [18]. For any set $\mathcal{A} \subseteq \mathbb{R}^p$, its Gaussian width is given by $w(\mathcal{A}) = \mathbb{E}\left[\sup_{\mathbf{u} \in \mathcal{A}} \langle \mathbf{u}, \mathbf{g} \rangle\right]$, where $\mathbf{g} \sim \mathcal{N}(\mathbf{0}, \mathbf{I}_{p \times p})$ is a standard Gaussian random vector. In the analysis, the set $\mathcal{A}$ of our interests typically relies on the structure of $\boldsymbol{\theta}^*$. Previously Gaussian width has been applied to statistical analyses for various problems [11, 6, 39], and recent works [34, 13] show that Gaussian width is computable for many structures. For the rest of the paper, we use $C, C_0, C_1$ and so on to denote universal constants, which are different from context to context.

## 3.1 Estimation of Coefficient Vector

In this subsection, we focus on estimating $\boldsymbol{\theta}^*$, i.e., Step 3 of Algorithm 1, using GDS of the form,

$$\hat{\boldsymbol{\theta}} = \operatorname*{argmin}_{\boldsymbol{\theta} \in \mathbb{R}^p} \|\boldsymbol{\theta}\| \quad \text{s.t.} \quad \left\| \frac{1}{n} \sum_{i=1}^{n} \mathbf{X}_i^T \boldsymbol{\Sigma}^{-1} (\mathbf{X}_i \boldsymbol{\theta} - \mathbf{y}_i) \right\|_* \leq \gamma_n \,, \tag{11}$$

where $\boldsymbol{\Sigma}$ is an arbitrary but fixed input noise covariance matrix. The following lemma shows a *deterministic* error bound for $\hat{\boldsymbol{\theta}}$ under the RE condition and admissible $\gamma_n$ defined in (9) and (10).

**Lemma 2** *Suppose the RE condition (9) is satisfied by $\mathbf{X}_1, \ldots, \mathbf{X}_n$ and $\boldsymbol{\Sigma}$ with $\alpha > 0$ for the set $\mathcal{A}(\boldsymbol{\theta}^*) = \operatorname{cone}\{\mathbf{v} \mid \|\boldsymbol{\theta}^* + \mathbf{v}\| \leq \|\boldsymbol{\theta}^*\|\} \cap \mathbb{S}^{p-1}$. If $\gamma_n$ is admissible, $\hat{\boldsymbol{\theta}}$ in (11) satisfies*

$$\left\| \hat{\boldsymbol{\theta}} - \boldsymbol{\theta}^* \right\|_2 \leq 2\Psi(\boldsymbol{\theta}^*) \cdot \frac{\gamma_n}{\alpha} \,, \tag{12}$$

*in which $\Psi(\boldsymbol{\theta}^*)$ is the restricted norm compatibility defined as $\Psi(\boldsymbol{\theta}^*) = \sup_{\mathbf{v} \in \mathcal{A}(\boldsymbol{\theta}^*)} \frac{\|\mathbf{v}\|}{\|\mathbf{v}\|_2}$.*

From Lemma 2, we can find that the $L_2$-norm error is mainly determined by three quantities–$\Psi(\boldsymbol{\theta}^*)$, $\gamma_n$ and $\alpha$. The restricted norm compatibility $\Psi(\boldsymbol{\theta}^*)$ purely hinges on the geometrical structure of $\boldsymbol{\theta}^*$ and $\|\cdot\|$, thus involving no randomness. On the contrary, $\gamma_n$ and $\alpha$ need to satisfy their own conditions, which are bound to deal with random $\mathbf{X}_i$ and $\boldsymbol{\eta}_i$. The set $\mathcal{A}(\boldsymbol{\theta}^*)$ involved in RE condition and restricted norm compatibility has relatively simple structure, which will favor the derivation of error bound for varieties of norms [13]. If RE condition fails to hold, i.e. $\alpha = 0$, the error bound is meaningless. Though the error is proportional to the user-specified $\gamma_n$, assigning arbitrarily small value to $\gamma_n$ may not be admissible. Hence, in order to further derive the recovery guarantees for GDS, we need to verify RE condition and find the smallest admissible value of $\gamma_n$.

**Restricted Eigenvalue Condition:** Firstly the following lemma characterizes the relation between the expectation and empirical mean of $\mathbf{X}^T \boldsymbol{\Sigma}^{-1} \mathbf{X}$.

**Lemma 3** *Given sub-Gaussian $\mathbf{X} \in \mathbb{R}^{m \times p}$ with its i.i.d. copies $\mathbf{X}_1, \ldots, \mathbf{X}_n$, and covariance $\boldsymbol{\Sigma} \in \mathbb{R}^{m \times m}$ with eigenvectors $\mathbf{u}_1, \ldots, \mathbf{u}_m$, let $\boldsymbol{\Gamma} = \mathbb{E}[\mathbf{X}^T \boldsymbol{\Sigma}^{-1} \mathbf{X}]$ and $\hat{\boldsymbol{\Gamma}} = \frac{1}{n} \sum_{i=1}^{n} \mathbf{X}_i^T \boldsymbol{\Sigma}^{-1} \mathbf{X}_i$. Define the set $\mathcal{A}_{\boldsymbol{\Gamma}_j}$ for $\mathcal{A} \subseteq \mathbb{S}^{p-1}$ and each $\boldsymbol{\Gamma}_j = \mathbb{E}[\mathbf{X}^T \mathbf{u}_j \mathbf{u}_j^T \mathbf{X}]$ as $\mathcal{A}_{\boldsymbol{\Gamma}_j} = \{\mathbf{v} \in \mathbb{S}^{p-1} \mid \boldsymbol{\Gamma}_j^{-\frac{1}{2}} \mathbf{v} \in \mathrm{cone}(\mathcal{A})\}$. If $n \geq C_1 \kappa^4 \cdot \max_j \{w^2(\mathcal{A}_{\boldsymbol{\Gamma}_j})\}$, with probability at least $1 - m \exp(-C_2 n/\kappa^4)$, we have*

$$\mathbf{v}^T \hat{\boldsymbol{\Gamma}} \mathbf{v} \geq \frac{1}{2} \mathbf{v}^T \boldsymbol{\Gamma} \mathbf{v}, \quad \forall \, \mathbf{v} \in \mathcal{A} \,. \tag{13}$$

Instead of $w(\mathcal{A}_{\boldsymbol{\Gamma}_j})$, ideally we want the condition above on $n$ to be characterized by $w(\mathcal{A})$, which can be easier to compute in general. The next lemma accomplishes this goal.

**Lemma 4** *Let $\kappa_0$ be the $\psi_2$-norm of standard Gaussian random vector and $\boldsymbol{\Gamma}_{\mathbf{u}} = \mathbb{E}[\mathbf{X}^T \mathbf{u} \mathbf{u}^T \mathbf{X}]$, where $\mathbf{u} \in \mathbb{S}^{m-1}$ is fixed. For $\mathcal{A}_{\boldsymbol{\Gamma}_{\mathbf{u}}}$ defined in Lemma 3, we have*

$$w(\mathcal{A}_{\boldsymbol{\Gamma}_{\mathbf{u}}}) \leq C \kappa_0 \sqrt{\mu_{\max}/\mu_{\min}} \cdot (w(\mathcal{A}) + 3) \,, \tag{14}$$

Lemma 4 implies that the Gaussian width $w(\mathcal{A}_{\boldsymbol{\Gamma}_j})$ appearing in Lemma 3 is of the same order as $w(\mathcal{A})$. Putting Lemma 3 and 4 together, we can obtain the RE condition for the analysis of GDS.

**Corollary 1** *Under the notations of Lemma 3 and 4, if $n \geq C_1 \kappa_0^2 \kappa^4 \cdot \frac{\mu_{\max}}{\mu_{\min}} \cdot (w(\mathcal{A}) + 3)^2$, then the following inequality holds for all $\mathbf{v} \in \mathcal{A} \subseteq \mathbb{S}^{p-1}$ with probability at least $1 - m \exp(-C_2 n/\kappa^4)$,*

$$\mathbf{v}^T \hat{\boldsymbol{\Gamma}} \mathbf{v} \geq \frac{\mu_{\min}}{2} \cdot \mathrm{Tr}(\boldsymbol{\Sigma}^{-1}) \tag{15}$$

**Admissible Tuning Parameter:** Finding the admissible $\gamma_n$ amounts to estimating the value of $\|\frac{1}{n} \sum_{i=1}^{n} \mathbf{X}_i^T \boldsymbol{\Sigma}^{-1} \boldsymbol{\eta}_i\|_*$ in (10), which involves random $\mathbf{X}_i$ and $\boldsymbol{\eta}_i$. The next lemma establishes a high-probability bound for this quantity, which can be viewed as the smallest "safe" choice of $\gamma_n$.

**Lemma 5** *Assume that $\mathbf{X}_i$ is sub-Gaussian and $\boldsymbol{\eta}_i \sim \mathcal{N}(\mathbf{0}, \boldsymbol{\Sigma}_*)$. The following inequality holds with probability at least $1 - \exp\left(-\frac{n\tau^2}{2}\right) - C_2 \exp\left(-\frac{C_1^2 w^2(\mathcal{B})}{4\rho^2}\right)$*

$$\left\| \frac{1}{n} \sum_{i=1}^{n} \mathbf{X}_i^T \boldsymbol{\Sigma}^{-1} \boldsymbol{\eta}_i \right\|_* \leq \frac{C \kappa \sqrt{\mu_{\max}}}{\sqrt{n}} \cdot \sqrt{\mathrm{Tr}\left(\boldsymbol{\Sigma}^{-1} \boldsymbol{\Sigma}_* \boldsymbol{\Sigma}^{-1}\right)} \cdot w(\mathcal{B}) \,, \tag{16}$$

*where $\mathcal{B}$ denotes the unit ball of norm $\|\cdot\|$, $\rho = \sup_{\mathbf{v} \in \mathcal{B}} \|\mathbf{v}\|_2$, and $\tau = \|\boldsymbol{\Sigma}^{-1} \boldsymbol{\Sigma}_*^{\frac{1}{2}}\|_F / \|\boldsymbol{\Sigma}^{-1} \boldsymbol{\Sigma}_*^{\frac{1}{2}}\|_2$.*

**Estimation Error of GDS:** Building on Corollary 1, Lemma 2 and 5, the theorem below characterizes the estimation of GDS for the multi-response linear model.

**Theorem 1** *Under the setting of Lemma 5, if $n \geq C_1 \kappa_0^2 \kappa^4 \cdot \frac{\mu_{\max}}{\mu_{\min}} \cdot (w(\mathcal{A}(\boldsymbol{\theta}^*)) + 3)^2$, and $\gamma_n$ is set to $C_2 \kappa \sqrt{\frac{\mu_{\max} \mathrm{Tr}(\boldsymbol{\Sigma}^{-1} \boldsymbol{\Sigma}_* \boldsymbol{\Sigma}^{-1})}{n}} \cdot w(\mathcal{B})$, the estimation error of $\hat{\boldsymbol{\theta}}$ given by (11) satisfies*

$$\|\hat{\boldsymbol{\theta}} - \boldsymbol{\theta}^*\|_2 \leq C \kappa \sqrt{\frac{\mu_{\max}}{\mu_{\min}^2}} \cdot \frac{\sqrt{\mathrm{Tr}\left(\boldsymbol{\Sigma}^{-1} \boldsymbol{\Sigma}_* \boldsymbol{\Sigma}^{-1}\right)}}{\mathrm{Tr}\left(\boldsymbol{\Sigma}^{-1}\right)} \cdot \frac{\Psi(\boldsymbol{\theta}^*) \cdot w(\mathcal{B})}{\sqrt{n}} \,, \tag{17}$$

*with probability at least* $1 - m \exp\left(-\frac{C_3 n}{\kappa^4}\right) - \exp\left(-\frac{n\tau^2}{2}\right) - C_4 \exp\left(-\frac{C_5^2 w^2(\mathcal{B})}{4\rho^2}\right)$.

**Remark:** We can see from the theorem above that the noise covariance $\mathbf{\Sigma}$ input to GDS plays a role in the error bound through the multiplicative factor $\xi(\mathbf{\Sigma}) = \sqrt{\text{Tr}\left(\mathbf{\Sigma}^{-1}\mathbf{\Sigma}_*\mathbf{\Sigma}^{-1}\right)}/\text{Tr}\left(\mathbf{\Sigma}^{-1}\right)$. By taking the derivative of $\xi^2(\mathbf{\Sigma})$ w.r.t. $\mathbf{\Sigma}^{-1}$ and setting it to $\mathbf{0}$, we have

$$\frac{\partial \xi^2(\mathbf{\Sigma})}{\partial \mathbf{\Sigma}^{-1}} = \frac{2\,\text{Tr}^2\left(\mathbf{\Sigma}^{-1}\right)\mathbf{\Sigma}_*\mathbf{\Sigma}^{-1} - 2\,\text{Tr}\left(\mathbf{\Sigma}^{-1}\right)\text{Tr}\left(\mathbf{\Sigma}^{-1}\mathbf{\Sigma}_*\mathbf{\Sigma}^{-1}\right)\cdot\mathbf{I}_{m\times m}}{\text{Tr}^4\left(\mathbf{\Sigma}^{-1}\right)} = \mathbf{0}$$

Then we can verify that $\mathbf{\Sigma} = \mathbf{\Sigma}_*$ is the solution to the equation above, and thus is the minimizer of $\xi(\mathbf{\Sigma})$ with $\xi(\mathbf{\Sigma}_*) = 1/\sqrt{\text{Tr}(\mathbf{\Sigma}_*^{-1})}$. This calculation confirms that multi-response regression could benefit from taking into account the noise covariance, and the best performance is achieved when $\mathbf{\Sigma}_*$ is known. If we perform ordinary GDS by setting $\mathbf{\Sigma} = \mathbf{I}_{m\times m}$, then $\xi(\mathbf{\Sigma}) = 1/\sqrt{m}$. Therefore using $\mathbf{\Sigma}_*$ will reduce the error by a factor of $\sqrt{m/\text{Tr}(\mathbf{\Sigma}_*^{-1})}$, compared with ordinary GDS.

One simple structure of $\boldsymbol{\theta}^*$ to consider for Theorem 1 is the sparsity encoded by $L_1$ norm. Given $s$-sparse $\boldsymbol{\theta}^*$, it follows from previous results [31, 11] that $\Psi(\boldsymbol{\theta}^*) = O(\sqrt{s})$, $w(\mathcal{A}(\boldsymbol{\theta}^*)) = O(\sqrt{s\log p})$ and $w(\mathcal{B}) = O(\sqrt{\log p})$. Therefore if $n \geq O(s\log p)$, then with high probability we have

$$\|\hat{\boldsymbol{\theta}} - \boldsymbol{\theta}^*\|_2 \leq O\left(\xi(\mathbf{\Sigma})\cdot\sqrt{\frac{s\log p}{n}}\right) \tag{18}$$

**Implications for Simple Linear Models:** Our general result in multi-response scenario implies some existing results for simple linear models. If we set $n = 1$ and $\mathbf{\Sigma} = \mathbf{\Sigma}_* = \mathbf{I}_{m\times m}$, i.e., only one data point $(\mathbf{X}, \mathbf{y})$ is observed and the noise is independent for each response, the GDS is reduced to

$$\hat{\boldsymbol{\theta}}_{\text{sg}} = \underset{\boldsymbol{\theta}\in\mathbb{R}^p}{\arg\min} \|\boldsymbol{\theta}\| \quad \text{s.t.} \quad \left\|\mathbf{X}^T(\mathbf{X}\boldsymbol{\theta} - \mathbf{y})\right\|_* \leq \gamma, \tag{19}$$

which exactly matches that in [12]. To bound its estimation error, we need $\mathbf{X}$ to be more structured beyond the sub-Gaussianity. Essentially we consider the model of $\mathbf{X}$ in Lemma 1, where rows of $\tilde{\mathbf{X}}$ are additionally assumed to be identical. For such $\mathbf{X}$, a specialized RE condition is as follows.

**Lemma 6** *Assume $\mathbf{X}$ is defined as in Lemma 1 such that $\mathbf{X} = \mathbf{\Xi}^{\frac{1}{2}}\tilde{\mathbf{X}}\mathbf{\Lambda}^{\frac{1}{2}}$, and rows of $\tilde{\mathbf{X}}$ are i.i.d. with $\|\tilde{\mathbf{x}}_j\| \leq \tilde{\kappa}$. If $mn \geq C_1\kappa_0^2\tilde{\kappa}^4 \cdot \frac{\lambda_{\max}(\mathbf{\Xi})\lambda_{\max}(\mathbf{\Lambda})}{\lambda_{\min}(\mathbf{\Xi})\lambda_{\min}(\mathbf{\Lambda})} \cdot (w(\mathcal{A}) + 3)^2$, with probability at least $1 - \exp(-C_2 mn/\tilde{\kappa}^4)$, the following inequality is satisfied by all $\mathbf{v} \in \mathcal{A} \subseteq \mathbb{S}^{p-1}$,*

$$\mathbf{v}^T\hat{\mathbf{\Gamma}}\mathbf{v} \geq \frac{m}{2}\cdot\lambda_{\min}\left(\mathbf{\Xi}^{\frac{1}{2}}\mathbf{\Sigma}^{-1}\mathbf{\Xi}^{\frac{1}{2}}\right)\cdot\lambda_{\min}(\mathbf{\Lambda}). \tag{20}$$

**Remark:** Lemma 6 characterizes the RE condition for a class of specifically structured design $\mathbf{X}$. If we specialize the general RE condition in Corollary 1 for this setting, $\mathbf{X} = \mathbf{\Xi}^{\frac{1}{2}}\tilde{\mathbf{X}}\mathbf{\Lambda}^{\frac{1}{2}}$, it becomes

$$n \geq C_1\kappa_0^2\tilde{\kappa}^4\frac{\lambda_{\max}(\mathbf{\Xi})\lambda_{\max}(\mathbf{\Lambda})}{\lambda_{\min}(\mathbf{\Xi})\lambda_{\min}(\mathbf{\Lambda})}(w(\mathcal{A}) + 3)^2 \xrightarrow[\quad\quad\quad\quad\quad]{\substack{\text{with probability } 1-\\ m\exp(-C_2 n/\tilde{\kappa}^4)}} \mathbf{v}^T\hat{\mathbf{\Gamma}}\mathbf{v} \geq \frac{\lambda_{\min}(\mathbf{\Xi})\lambda_{\min}(\mathbf{\Lambda})}{2}\text{Tr}(\mathbf{\Sigma}^{-1})$$

Comparing the general result above with Lemma 6, there are two striking differences. Firstly, Lemma 6 requires the same sample size of $mn$ rather than $n$, which improves the general one. Secondly, (20) holds with much higher probability $1 - \exp(-C_2 mn/\tilde{\kappa}^4)$ instead of $1 - m\exp(-C_2 n/\tilde{\kappa}^4)$.

Given this specialized RE condition, we have the recovery guarantees of GDS for simple linear models, which encompass the settings discussed in [6, 12] as special cases.

**Corollary 2** *Suppose $\mathbf{y} = \mathbf{X}\boldsymbol{\theta}^* + \boldsymbol{\eta} \in \mathbb{R}^m$, where $\mathbf{X}$ is described as in Lemma 6, and $\boldsymbol{\eta} \sim \mathcal{N}(\mathbf{0}, \mathbf{I})$. With probability at least $1 - \exp\left(-\frac{m}{2}\right) - C_2\exp\left(-\frac{C_1^2 w^2(\mathcal{B})}{4\rho^2}\right) - \exp\left(-C_3 m/\tilde{\kappa}^4\right)$, $\hat{\boldsymbol{\theta}}_{sg}$ satisfies*

$$\left\|\hat{\boldsymbol{\theta}}_{sg} - \boldsymbol{\theta}^*\right\|_2 \leq C\tilde{\kappa}\cdot\sqrt{\frac{\lambda_{\max}(\mathbf{\Xi})\lambda_{\max}(\mathbf{\Lambda})}{\lambda_{\min}^2(\mathbf{\Xi})\lambda_{\min}^2(\mathbf{\Lambda})}}\cdot\frac{\Psi(\boldsymbol{\theta}^*)\cdot w(\mathcal{B})}{\sqrt{m}}, \tag{21}$$

## 3.2 Estimation of Noise Covariance

In this subsection, we consider the estimation of noise covariance $\boldsymbol{\Sigma}_*$ given an arbitrary parameter vector $\boldsymbol{\theta}$. When $m$ is small, we estimate $\boldsymbol{\Sigma}_*$ by simply using the sample covariance

$$\hat{\boldsymbol{\Sigma}} = \frac{1}{n} \sum_{i=1}^{n} \left(\mathbf{y}_i - \mathbf{X}_i \boldsymbol{\theta}\right) \left(\mathbf{y}_i - \mathbf{X}_i \boldsymbol{\theta}\right)^T . \tag{22}$$

Theorem 2 reveals the relation between $\hat{\boldsymbol{\Sigma}}$ and $\boldsymbol{\Sigma}_*$, which is sufficient for our AltEst analysis.

**Theorem 2** *If* $n \geq C^4 m \cdot \max\left\{ 4\left(\kappa_0 + \kappa \sqrt{\frac{\mu_{\max}}{\lambda_{\min}(\boldsymbol{\Sigma}_*)}} \|\boldsymbol{\theta}^* - \boldsymbol{\theta}\|_2\right)^4 , \kappa^4 \left(\frac{\lambda_{\max}(\boldsymbol{\Sigma}_*)\mu_{\max}}{\lambda_{\min}(\boldsymbol{\Sigma}_*)\mu_{\min}}\right)^2 \right\}$ *and*

$\mathbf{X}_i$ *is sub-Gaussian, with probability at least* $1 - 2\exp(-C_1 m)$, $\hat{\boldsymbol{\Sigma}}$ *given by* (22) *satisfies*

$$\lambda_{\max}\left(\boldsymbol{\Sigma}_*^{-\frac{1}{2}} \hat{\boldsymbol{\Sigma}} \boldsymbol{\Sigma}_*^{-\frac{1}{2}}\right) \leq 1 + C^2 \kappa_0^2 \sqrt{m/n} + \frac{2\mu_{\max}}{\lambda_{\min}(\boldsymbol{\Sigma}_*)} \|\boldsymbol{\theta}^* - \boldsymbol{\theta}\|_2^2 \tag{23}$$

$$\lambda_{\min}\left(\boldsymbol{\Sigma}_*^{-\frac{1}{2}} \hat{\boldsymbol{\Sigma}} \boldsymbol{\Sigma}_*^{-\frac{1}{2}}\right) \geq 1 - C^2 \kappa_0^2 \sqrt{m/n} \tag{24}$$

**Remark:** If $\hat{\boldsymbol{\Sigma}} = \boldsymbol{\Sigma}_*$, then $\lambda_{\max}(\boldsymbol{\Sigma}_*^{-\frac{1}{2}} \hat{\boldsymbol{\Sigma}} \boldsymbol{\Sigma}_*^{-\frac{1}{2}}) = \lambda_{\min}(\boldsymbol{\Sigma}_*^{-\frac{1}{2}} \hat{\boldsymbol{\Sigma}} \boldsymbol{\Sigma}_*^{-\frac{1}{2}}) = 1$. Hence $\hat{\boldsymbol{\Sigma}}$ is nearly equal to $\boldsymbol{\Sigma}_*$ when the upper and lower bounds (23) (24) are close to 1. We would like to point out that there is nothing specific to the particular form of estimator (22), which makes AltEst work. Similar results can be obtained for other methods that estimate the inverse covariance matrix $\boldsymbol{\Sigma}_*^{-1}$ instead of $\boldsymbol{\Sigma}_*$. For instance, when $m < n$ and $\boldsymbol{\Sigma}_*^{-1}$ is sparse, we can replace (22) with GLasso [16] or CLIME [9], and AltEst only requires the counterparts of (23) and (24) in order to work.

## 3.3 Error Bound for Alternating Estimation

Section 3.1 shows that the noise covariance in GDS affects the error bound by the factor $\xi(\boldsymbol{\Sigma})$. In order to bound the error of $\hat{\boldsymbol{\theta}}_T$ given by AltEst, we need to further quantify how $\boldsymbol{\theta}$ affects $\xi(\hat{\boldsymbol{\Sigma}})$.

**Lemma 7** *If* $\hat{\boldsymbol{\Sigma}}$ *is given as* (22) *and the condition in Theorem 2 holds, then the inequality below holds with probability at least* $1 - 2\exp(-C_1 m)$,

$$\xi\left(\hat{\boldsymbol{\Sigma}}\right) \leq \xi\left(\boldsymbol{\Sigma}_*\right) \cdot \left(1 + 2C\kappa_0 \left(\frac{m}{n}\right)^{\frac{1}{4}} + 2\sqrt{\frac{\mu_{\max}}{\lambda_{\min}(\boldsymbol{\Sigma}_*)}} \|\boldsymbol{\theta}^* - \boldsymbol{\theta}\|_2\right) \tag{25}$$

Based on Lemma 7, the following theorem provides the error bound for $\hat{\boldsymbol{\theta}}_T$ given by Algorithm 1.

**Theorem 3** *Let* $e_{orc} = C_1 \kappa \sqrt{\frac{\mu_{\max}}{\mu_{\min}^2}} \frac{\xi(\boldsymbol{\Sigma}_*) \cdot \Psi(\boldsymbol{\theta}^*) w(\mathcal{B})}{\sqrt{n}}$ *and* $e_{min} = e_{orc} \cdot \frac{1 + 2C\kappa_0 \left(\frac{m}{n}\right)^{\frac{1}{4}}}{1 - 2e_{orc}\sqrt{\frac{\mu_{\max}}{\lambda_{\min}(\boldsymbol{\Sigma}_*)}}}$. *If* $n \geq C^4 m \cdot$

$\max\left\{ 4\left(\kappa_0 + \frac{C_1}{C^2}\sqrt{\frac{\lambda_{\min}(\boldsymbol{\Sigma}_*)}{\lambda_{\max}^2(\boldsymbol{\Sigma}_*)}} \frac{\Psi(\boldsymbol{\theta}^*) w(\mathcal{B})}{m}\right)^4 , \kappa^4 \left(\frac{\lambda_{\max}(\boldsymbol{\Sigma}_*)\mu_{\max}}{\lambda_{\min}(\boldsymbol{\Sigma}_*)\mu_{\min}}\right)^2 , \left(\frac{2C_1\kappa\mu_{\max}}{C^2\mu_{\min}} \cdot \frac{\xi(\boldsymbol{\Sigma}_*)\Psi(\boldsymbol{\theta}^*) w(\mathcal{B})}{\sqrt{m \cdot \lambda_{\min}(\boldsymbol{\Sigma}_*)}}\right)^2 \right\}$

*and also satisfies the condition in Theorem 1, with high probability, the iterate* $\hat{\boldsymbol{\theta}}_T$ *returned by Algorithm 1 satisfies*

$$\left\|\hat{\boldsymbol{\theta}}_T - \boldsymbol{\theta}^*\right\|_2 \leq e_{min} + \left(2e_{orc}\sqrt{\frac{\mu_{\max}}{\lambda_{\min}(\boldsymbol{\Sigma}_*)}}\right)^{T-1} \cdot \left(\left\|\hat{\boldsymbol{\theta}}_1 - \boldsymbol{\theta}^*\right\|_2 - e_{min}\right) \tag{26}$$

**Remark:** The three lower bounds for $n$ inside curly braces correspond to three intuitive requirements. The first one guarantees that the covariance estimation is accurate enough, and the other two respectively ensure that the initial error of $\hat{\boldsymbol{\theta}}_1$ and $e_{orc}$ are reasonably small , such that the subsequent errors can contract linearly. $e_{orc}$ is the estimation error incurred by the following oracle estimator,

$$\hat{\boldsymbol{\theta}}_{orc} = \underset{\boldsymbol{\theta} \in \mathbb{R}^p}{\operatorname{argmin}} \|\boldsymbol{\theta}\| \quad \text{s.t.} \quad \left\| \frac{1}{n} \sum_{i=1}^{n} \mathbf{X}_i^T \boldsymbol{\Sigma}_*^{-1} (\mathbf{X}_i \boldsymbol{\theta} - \mathbf{y}_i) \right\|_* \leq \gamma_n , \tag{27}$$

which is impossible to implement in practice. On the other hand, $e_{min}$ is the minimum achievable error, which has an extra multiplicative factor compared with $e_{orc}$. The numerator of the factor compensates

for the error of estimated noise covariance provided that $\boldsymbol{\theta} = \boldsymbol{\theta}^*$ is plugged in (22), which merely depends on sample size. Since having $\boldsymbol{\theta} = \boldsymbol{\theta}^*$ is also unrealistic for (22), the denominator further accounts for the ballpark difference between $\boldsymbol{\theta}$ and $\boldsymbol{\theta}^*$. As we remark after Theorem 1, if we perform ordinary GDS with $\boldsymbol{\Sigma}$ set to $\mathbf{I}_{m \times m}$ in (11), its error bound $e_{\mathrm{odn}}$ satisfies $e_{\mathrm{odn}} = e_{\mathrm{orc}}\sqrt{\mathrm{Tr}(\boldsymbol{\Sigma}_*^{-1})/m}$. Note that this factor $\sqrt{\mathrm{Tr}(\boldsymbol{\Sigma}_*^{-1})/m}$ is independent of $n$, whereas $e_{\min}$ will approach $e_{\mathrm{orc}}$ with increasing $n$ as the factor between them converges to one.

# 4   Experiments

In this section, we present some experimental results to support our theoretical analysis. Specifically we focus on the sparse structure of $\boldsymbol{\theta}^*$ captured by $L_1$ norm. Throughout the experiment, we fix problem dimension $p = 500$, sparsity level of $\boldsymbol{\theta}^*$ $s = 20$, and number of iterations for AltEst $T = 5$. Entries of design $\mathbf{X}$ is generated by i.i.d. standard Gaussians, and $\boldsymbol{\theta}^* = [\underbrace{1, \ldots, 1}_{10}, \underbrace{-1, \ldots, -1}_{10}, \underbrace{0, \ldots, 0}_{480}]^T$. $\boldsymbol{\Sigma}_*$ is given as a block diagonal matrix with blocks $\boldsymbol{\Sigma}' = \begin{bmatrix} 1 & a \\ a & 1 \end{bmatrix}$ replicated along diagonal, and number of responses $m$ is assumed to be even. All plots are obtained by averaging 100 trials. In the first set of experiments, we set $a = 0.8$, $m = 10$ and investigate the error of $\hat{\boldsymbol{\theta}}_t$ as $n$ varies from 40 to 90. We run AltEst (with and without resampling), the oracle GDS, and the ordinary GDS with $\boldsymbol{\Sigma} = \mathbf{I}$. The results are given in Figure 1.

For the second experiment, we fix the product $mn \approx 500$, and let $m = 2, 4, \ldots, 10$. For our choice of $\boldsymbol{\Sigma}_*$, the error incurred by oracle GDS $e_{\mathrm{orc}}$ is the same for every $m$. We compare AltEst with both oracle and ordinary GDS, and the result is shown in Figure 2(a) and 2(b).

In the third experiment, we test AltEst under different covariance matrices $\boldsymbol{\Sigma}_*$ by varying $a$ from 0.5 to 0.9. $m$ is set to 10 and sample size $n$ is 90. We also compare AltEst against both oracle and ordinary GDS, and the errors are reported in Figure 2(c) and 2(d).

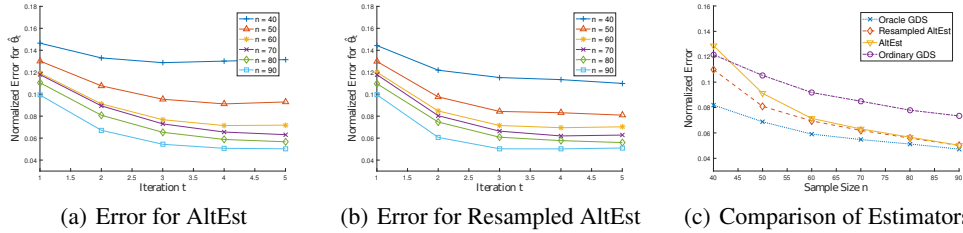

(a) Error for AltEst     (b) Error for Resampled AltEst     (c) Comparison of Estimators

Figure 1: (a) When $n = 40$, AltEst is not quite stable due to the large initial error and poor quality of estimated covariance. Then the errors start to decrease for $n \geq 50$. (b) Resampld AltEst does benefit from fresh samples, and its error is slightly smaller than AltEst as well as more stable when $n$ is small. (c) Oracle GDS outperforms the others, but the performance of AltEst is also competitive. Ordinary GDS is unable to utilize the noise correlation, thus resulting in relatively large error. By comparing the two implementations of AltEst, we can see that resampled AltEst yields smaller error especially when data is inadequate, but their errors are very close if $n$ is suitably large.

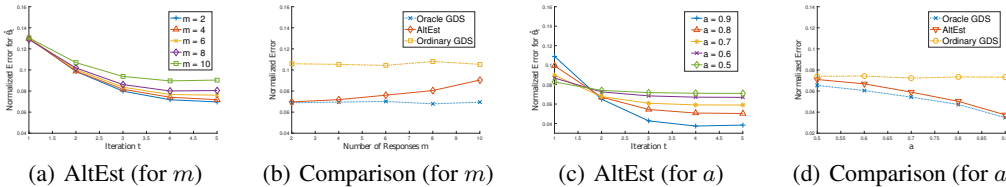

(a) AltEst (for $m$)     (b) Comparison (for $m$)     (c) AltEst (for $a$)     (d) Comparison (for $a$)

Figure 2: (a) Larger error comes with bigger $m$, which confirms that $e_{\min}$ is increasing along with $m$ when $mn$ is fixed. (b) The plots for oracle and ordinary GDS imply that $e_{\mathrm{orc}}$ and $e_{\mathrm{odn}}$ remain unchanged, which matches the error bounds in Theorem 1. Though $e_{\min}$ increases, AltEst still outperform the ordinary GDS by a margin. (c) The error goes down when the true noise covariance becomes closer to singular, which is expected in view of Theorem 3. (d) $e_{\mathrm{orc}}$ also decreases as $a$ gets larger, and the gap between $e_{\min}$ and $e_{\mathrm{odn}}$ widens. The definition of $e_{\min}$ in Theorem 3 indicates that the ratio between $e_{\min}$ and $e_{\mathrm{orc}}$ is almost a constant because both $n$ and $m$ are fixed. Here we observe that all the ratios at different $a$ are between 1.05 and 1.1, which supports the theoretical results. Also, Theorem 1 suggests that $e_{\mathrm{odn}}$ does not change as $\boldsymbol{\Sigma}_*$ varies, which is verified here.

## 5 Conclusions

In this paper, we propose an alternating estimation (AltEst) procedure for solving the multi-response linear models in high dimension. Our framework is based on the generalized Dantzig selector (GDS) and allows for general structures of the parameter vector, whose recovery guarantees are simply determined by a few geometric measures. Also, by leveraging the noise correlation among responses, AltEst can achieve significantly smaller estimation error than ignoring the noise structure. With moderate sample size and the resampling assumption, we show that the estimation error will converge linearly to a minimal achievable error, which is comparable to the one incurred by the oracle estimator. In the experiment, we demonstrate the numerical superiority of AltEst over the vanilla GDS, and it is also suggested that the resampled version of AltEst give little benefit in practice and we should better use all data in every iteration.

## Acknowledgements

The research was supported by NSF grants IIS-1563950, IIS-1447566, IIS-1447574, IIS-1422557, CCF-1451986, CNS- 1314560, IIS-0953274, IIS-1029711, NASA grant NNX12AQ39A, and gifts from Adobe, IBM, and Yahoo.

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
