[Supplementary Material · suppl_camera.pdf]

# Supplementary Material to Alternating Estimation for Structured High-Dimensional Multi-Response Models

**Sheng Chen**  **Arindam Banerjee**
Dept. of Computer Science & Engineering
University of Minnesota, Twin Cities
{shengc,banerjee}@cs.umn.edu

## 1 Preliminaries

In this section, we provide some background knowledge and lemmas, which is needed in our proofs. For the sake of convenience, $C$, $C_0$, $c$, $c_0$ and so on are reserved for absolute constants.

### 1.1 Sub-Gaussian Random Variable/Vector

A random variable $x$ is sub-Gaussian if the $\psi_2$-norm defined below is finite

$$\|x\|_{\psi_2} = \sup_{q \geq 1} \frac{\mathbb{E}|x|^q}{\sqrt{q}} < +\infty \tag{S.1}$$

A random vector $\mathbf{x} \in \mathbb{R}^p$ is sub-Gaussian if $\langle \mathbf{x}, \mathbf{u} \rangle$ is sub-Gaussian for any $\mathbf{u} \in \mathbb{R}^p$, and $\|\mathbf{x}\|_{\psi_2} = \sup_{\mathbf{u} \in \mathbb{R}^p} \|\langle \mathbf{x}, \mathbf{u} \rangle\|_{\psi_2}$. A complete introduction can be found in [6]. Here we list some of the well-known properties of sub-Gaussian random variables/vectors, which are extracted from [6].

**Proposition A (Sub-Gaussian Tail)** *A random variable $x$ satisfies the following inequality iff* $\|x\|_{\psi_2} \leq \kappa$,

$$\mathbb{P}\left(|x| > \epsilon\right) \leq e \cdot \exp\left(-\frac{C\epsilon^2}{\kappa^2}\right) , \tag{S.2}$$

*where $C$ is a absolute constant.*

**Proposition B** *If $x_1, x_2, \ldots, x_n$ are independent centered sub-Gaussian random variables, then $\sum_i x_i$ is also a centered sub-Gaussian random variable with*

$$\left\|\sum_{i=1}^{n} x_i\right\|_{\psi_2}^2 \leq C^2 \sum_{i=1}^{n} \|x_i\|_{\psi_2}^2 , \tag{S.3}$$

*where $C$ is an absolute constant.*

**Proposition C** *If $x_1, x_2, \ldots, x_n$ are independent centered sub-Gaussian random variables (not necessarily identical), then $\mathbf{x} = [x_1, \ldots, x_n]^T$ is a centered sub-Gaussian random vector with*

$$\|\mathbf{x}\|_{\psi_2} \leq C \max_{1 \leq i \leq n} \|x_i\|_{\psi_2} , \tag{S.4}$$

*where $C$ is an absolute constant.*

Essentially Proposition C can be shown using the definition of sub-Gaussian vector and Proposition B, which we generalize to independent sub-Gaussian vectors as follows.

**Lemma A** *If* $\mathbf{x}_1, \mathbf{x}_2, \ldots, \mathbf{x}_n$ *are all $m$-dimensional independent centered sub-Gaussian random vectors, then* $\mathbf{x} = [\mathbf{x}_1^T, \ldots, \mathbf{x}_n^T]^T \in \mathbb{R}^{mn}$ *is also a centered sub-Gaussian random vector with*

$$\|\|\mathbf{x}\|\|_{\psi_2} \leq C \max_{1 \leq i \leq n} \|\|\mathbf{x}_i\|\|_{\psi_2} , \tag{S.5}$$

*where $C$ is an absolute constant.*

*Proof:* Define $\mathbf{a} = [\mathbf{a}_1^T, \mathbf{a}_2^T, \ldots, \mathbf{a}_n^T]^T \in \mathbb{S}^{mn-1}$, where each $\mathbf{a}_i$ is $m$-dimensional. We have

$$\|\|\langle \mathbf{x}, \mathbf{a}\rangle\|\|_{\psi_2} = \left\|\left\|\sum_{i=1}^{n} \langle \mathbf{x}_i, \mathbf{a}_i\rangle\right\|\right\|_{\psi_2} \leq \sqrt{C^2 \sum_{i=1}^{n} \|\|\langle \mathbf{x}_i, \mathbf{a}_i\rangle\|\|_{\psi_2}^2} \leq \sqrt{C^2 \sum_{i=1}^{n} \|\mathbf{a}_i\|_2^2 \|\|\mathbf{x}_i\|\|_{\psi_2}^2}$$

$$\leq \sqrt{C^2 \sum_{i=1}^{n} \|\mathbf{a}_i\|_2^2} \cdot \max_{1 \leq i \leq n} \|\|\mathbf{x}_i\|\|_{\psi_2} = C \max_{1 \leq i \leq n} \|\|\mathbf{x}_i\|\|_{\psi_2} ,$$

where we use Proposition B for the first inequality. Based on the definition of sub-Gaussian random vector, we complete the proof. ∎

## 1.2 Generic Chaining and Gaussian Width

One important tool that we use in our probabilistic argument is *generic chaining* [4, 5], which is powerful for bounding the suprema of stochastic processes. Suppose $\{Z_\mathbf{t}\}_{\mathbf{t} \in \mathcal{T}}$ is a centered stochastic process, where each $Z_t$ is a centered random variable. We assume the index set $\mathcal{T}$ is endowed with some metric (distance function) $s(\cdot, \cdot)$. A key notion in generic chaining is $\gamma_2$-functional $\gamma_2(\mathcal{T}, s)$, which is defined for the metric space $(\mathcal{T}, s)$. One can think of $\gamma_2$-functional as a measure of the size of set $\mathcal{T}$ w.r.t. metric $s$. For self-containedness, we give the expression of $\gamma_2(\mathcal{T}, s)$.

$$\gamma_2(\mathcal{T}, s) = \inf_{\{\mathcal{P}_n\}} \sup_{\mathbf{t} \in \mathcal{T}} \sum_{n \geq 0} 2^{n/2} \cdot \operatorname{diam}(\mathcal{P}_n(\mathbf{t}), s) , \tag{S.6}$$

where $\{\mathcal{P}_n\}_{n=0}^{\infty} = \{\mathcal{P}_0, \mathcal{P}_1, \ldots, \mathcal{P}_n, \ldots\}$ is a sequence of partitions for $\mathcal{T}$, which satisfy that $|\mathcal{P}_0| = 1$, $|\mathcal{P}_n| \leq 2^{2^n}$ for $n \geq 1$, and that $\mathcal{P}_{n+1}$ is a finer partition than $\mathcal{P}_n$, i.e., every $\mathcal{Q} \in \mathcal{P}_{n+1}$ is a subset of some $\mathcal{Q}' \in \mathcal{P}_n$. $\mathcal{P}_n(\mathbf{t})$ denotes the subset of $\mathcal{T}$ that contains $\mathbf{t}$ in the $n$-th partition, and $\operatorname{diam}(\mathcal{P}_n(\mathbf{t}), s)$ measures the diameter of $\mathcal{P}_n(\mathbf{t})$ w.r.t. metric $s(\cdot, \cdot)$. Note that $\gamma_2$-functional is a purely geometric concept, which involves no probability. Given that $\gamma_2$-functional is fairly involved, we are not going to discuss any insights behind this definition, and refer interested readers to the introductory books [4, 5]. Based on its definition, we list a few straightforward properties of $\gamma_2$-functional here.

$$\gamma_2(\mathcal{T}, s_1) \leq \gamma_2(\mathcal{T}, s_2) \quad \text{if } s_1(\mathbf{u}, \mathbf{v}) \leq s_2(\mathbf{u}, \mathbf{v}), \forall \, \mathbf{u}, \mathbf{v} \in \mathcal{T} \tag{S.7}$$

$$\gamma_2(\mathcal{T}, \beta s) = \beta \cdot \gamma_2(\mathcal{T}, s) \quad \text{for any } \beta > 0 . \tag{S.8}$$

$$\gamma_2(\mathcal{T}_1, s_1) = \gamma_2(\mathcal{T}_2, s_2) \quad \text{if } \exists \text{ a global isometry between } (\mathcal{T}_1, s_1) \text{ and } (\mathcal{T}_2, s_2) \tag{S.9}$$

The following lemma concerned with the suprema of $\{Z_\mathbf{t}\}$ combines Theorem 2.2.22 and 2.2.27 from [5].

**Lemma B** *Given metric space $(\mathcal{T}, s)$, if the associated centered stochastic process $\{Z_\mathbf{t}\}_{\mathbf{t} \in \mathcal{T}}$ satisfies the condition*

$$\mathbb{P}\left(|Z_\mathbf{u} - Z_\mathbf{v}| \geq \epsilon\right) \leq C_0 \exp\left(-\frac{C_1 \epsilon^2}{s^2(\mathbf{u}, \mathbf{v})}\right), \quad \forall \, \mathbf{u}, \mathbf{v} \in \mathcal{T} , \tag{S.10}$$

*then the following inequalities hold*

$$\mathbb{E}\left[\sup_{\mathbf{t} \in \mathcal{T}} Z_\mathbf{t}\right] \leq C_2 \gamma_2(\mathcal{T}, s) , \tag{S.11}$$

$$\mathbb{P}\left(\sup_{\mathbf{u}, \mathbf{v} \in \mathcal{T}} |Z_\mathbf{u} - Z_\mathbf{v}| \geq C_3 \left(\gamma_2(\mathcal{T}, s) + \epsilon \cdot \operatorname{diam}(\mathcal{T}, s)\right)\right) \leq C_4 \exp\left(-\epsilon^2\right) , \tag{S.12}$$

*where $C_0, C_1, C_2, C_3$ and $C_4$ are all absolute constants.*

Another useful result based on generic chaining is the Theorem D in [2].

**Lemma C (Theorem D in [2])** *There exist absolute constants $C_1$, $C_2$ for which the following holds. Let $(\Omega, \mu)$ be a probability space on which $X$ is defined, and $X_1, \ldots, X_n$ be independent copies of $X$. Let set $\mathcal{H}$ be a subset of the unit sphere of $L_2(\mu)$, i.e., $\mathcal{H} \subseteq \mathbb{S}_{L_2} = \{h : \|h\|_{L_2} = \sqrt{\int_\Omega h^2(X) dX} = 1\}$, and assume that $\sup_{h \in \mathcal{H}} \|h\|_{\psi_2} \leq \kappa$. Then, for any $\beta > 0$ and $n \geq 1$ satisfying*

$$C_1 \kappa \gamma_2(\mathcal{H}, \|\cdot\|_{\psi_2}) \leq \beta \sqrt{n} , \tag{S.13}$$

*with probability at least $1 - \exp(-C_2 \beta^2 n / \kappa^4)$,*

$$\sup_{h \in \mathcal{H}} \left| \frac{1}{n} \sum_{i=1}^{n} h^2(X_i) - \mathbb{E}\left[h^2\right] \right| \leq \beta . \tag{S.14}$$

The suprema in both Lemma B and C are characterized in terms of $\gamma_2$-functional, which is not easily computable. In order to further bound the $\gamma_2$-functional, one needs the so-called *majorizing measures theorem* [3].

**Lemma D** *Given any Gaussian process $\{Y_\mathbf{t}\}_{\mathbf{t} \in \mathcal{T}}$, define $s(\mathbf{u}, \mathbf{v}) = \sqrt{\mathbb{E}|Y_\mathbf{u} - Y_\mathbf{v}|^2}$ for $\mathbf{u}, \mathbf{v} \in \mathcal{T}$. Then $\gamma_2(\mathcal{T}, s)$ can be upper bounded by*

$$\gamma_2(\mathcal{T}, s) \leq C_0 \mathbb{E}\left[\sup_{\mathbf{t} \in \mathcal{T}} Y_\mathbf{t}\right] , \tag{S.15}$$

*where $C_0$ is an absolute constant.*

We construct the simple Gaussian process $\{Y_\mathbf{t} = \langle \mathbf{t}, \mathbf{g} \rangle\}_{\mathbf{t} \in \mathcal{T}}$ for any $\mathcal{T} \subseteq \mathbb{R}^p$, where $\mathbf{g}$ is a standard Gaussian random vector. Hence $s(\mathbf{u}, \mathbf{v}) = \sqrt{\mathbb{E}|Y_\mathbf{u} - Y_\mathbf{v}|^2} = \sqrt{\mathbb{E}|\langle \mathbf{u} - \mathbf{v}, \mathbf{g} \rangle|^2} = \|\mathbf{u} - \mathbf{v}\|_2$. It follows from Lemma D that

$$\gamma_2\left(\mathcal{T}, \|\cdot\|_2\right) \leq C_0 \mathbb{E}\left[\sup_{\mathbf{t} \in \mathcal{T}} \langle \mathbf{t}, \mathbf{g} \rangle\right] = C_0 \cdot w(\mathcal{T}) , \tag{S.16}$$

which makes the connection between $\gamma_2$-functional and Gaussian width. One technique we utilize in our proof for bounding Gaussian width is as follows, which originates in [1].

**Lemma E (Lemma 2 in [1])** *Let $M > 4$, $\mathcal{A}_1, \cdots, \mathcal{A}_M \subset \mathbb{R}^p$, and $\mathcal{A} = \cup_m \mathcal{A}_m$. The Gaussian width of $\mathcal{A}$ satisfies*

$$w(\mathcal{A}) \leq \max_{1 \leq m \leq M} w(\mathcal{A}_m) + 2 \sup_{\mathbf{z} \in \mathcal{A}} \|\mathbf{z}\|_2 \sqrt{\log M} \tag{S.17}$$

## 1.3 Proof of Lemma 1

**Statement of Lemma 1:** *Assume that $\mathbf{X} \in \mathbb{R}^{m \times p}$ has dependent anisotropic rows such that $\mathbf{X} = \boldsymbol{\Xi}^{\frac{1}{2}} \tilde{\mathbf{X}} \boldsymbol{\Lambda}^{\frac{1}{2}}$, where $\boldsymbol{\Xi} \in \mathbb{R}^{m \times m}$ encodes the dependency between rows, $\tilde{\mathbf{X}} \in \mathbb{R}^{m \times p}$ has independent isotropic rows, and $\boldsymbol{\Lambda} \in \mathbb{R}^{p \times p}$ introduces the anisotropicity. In this setting, if each row of $\tilde{\mathbf{X}}$ satisfies $\|\tilde{\mathbf{x}}_i\|_{\psi_2} \leq \tilde{\kappa}$, then condition (7) and (8) hold with $\kappa = C\tilde{\kappa}$, $\mu_{\min} = \lambda_{\min}(\boldsymbol{\Xi})\lambda_{\min}(\boldsymbol{\Lambda})$, and $\mu_{\max} = \lambda_{\max}(\boldsymbol{\Xi})\lambda_{\max}(\boldsymbol{\Lambda})$.*

*Proof:* Let $\mathbf{w} = \boldsymbol{\Xi}^{\frac{1}{2}} \mathbf{u}$ for any $\mathbf{u} \in \mathbb{S}^{m-1}$, and we have

$$\boldsymbol{\Gamma}_\mathbf{u} = \mathbb{E}\left[\boldsymbol{\Lambda}^{\frac{1}{2}} \tilde{\mathbf{X}}^T \boldsymbol{\Xi}^{\frac{1}{2}} \mathbf{u} \mathbf{u}^T \boldsymbol{\Xi}^{\frac{1}{2}} \tilde{\mathbf{X}} \boldsymbol{\Lambda}^{\frac{1}{2}}\right]$$

$$= \mathbb{E}\left[\left[\boldsymbol{\Lambda}^{\frac{1}{2}} \tilde{\mathbf{x}}_1, \ldots, \boldsymbol{\Lambda}^{\frac{1}{2}} \tilde{\mathbf{x}}_m\right] \cdot \begin{bmatrix} w_1 \\ \vdots \\ w_m \end{bmatrix} \cdot [w_1, \ldots, w_m] \cdot \begin{bmatrix} \tilde{\mathbf{x}}_1^T \boldsymbol{\Lambda}^{\frac{1}{2}} \\ \vdots \\ \tilde{\mathbf{x}}_m^T \boldsymbol{\Lambda}^{\frac{1}{2}} \end{bmatrix}\right]$$

$$= \sum_{i=1}^{m} \sum_{j=1}^{m} w_i w_j \mathbb{E}\left[\boldsymbol{\Lambda}^{\frac{1}{2}} \tilde{\mathbf{x}}_i \tilde{\mathbf{x}}_j^T \boldsymbol{\Lambda}^{\frac{1}{2}}\right] = \sum_{i=1}^{m} w_i^2 \boldsymbol{\Lambda}^{\frac{1}{2}} \mathbb{E}\left[\tilde{\mathbf{x}}_i \tilde{\mathbf{x}}_i^T\right] \boldsymbol{\Lambda}^{\frac{1}{2}} = \left\|\boldsymbol{\Xi}^{\frac{1}{2}} \mathbf{u}\right\|_2^2 \cdot \boldsymbol{\Lambda}$$

It is clear that

$$\lambda_{\min}(\boldsymbol{\Xi}) \cdot \lambda_{\min}(\boldsymbol{\Lambda}) \le \lambda_{\min}(\boldsymbol{\Gamma_u}) \le \lambda_{\max}(\boldsymbol{\Gamma_u}) \le \lambda_{\max}(\boldsymbol{\Xi}) \cdot \lambda_{\max}(\boldsymbol{\Lambda}) \,,$$

which indicates that condition (8) holds. If $\left\| \|\tilde{\mathbf{x}}_i\| \right\|_{\psi_2} \le \tilde{\kappa}$, then

$$
\begin{aligned}
\|\mathbf{X}\|_{\psi_2} &= \sup_{\substack{\mathbf{v} \in \mathbb{S}^{p-1} \\ \mathbf{u} \in \mathbb{S}^{m-1}}} \left\| \left\| \mathbf{v}^T \boldsymbol{\Gamma_u}^{-\frac{1}{2}} \mathbf{X}^T \mathbf{u} \right\| \right\|_{\psi_2} = \sup_{\substack{\mathbf{v} \in \mathbb{S}^{p-1} \\ \mathbf{u} \in \mathbb{S}^{m-1}}} \left\| \left\| \frac{\mathbf{v}^T \boldsymbol{\Lambda}^{-\frac{1}{2}}}{\|\boldsymbol{\Xi}^{\frac{1}{2}} \mathbf{u}\|_2} \cdot \boldsymbol{\Lambda}^{\frac{1}{2}} \tilde{\mathbf{X}}^T \boldsymbol{\Xi}^{\frac{1}{2}} \mathbf{u} \right\| \right\|_{\psi_2} \\
&= \sup_{\substack{\mathbf{v} \in \mathbb{S}^{p-1} \\ \mathbf{u} \in \mathbb{S}^{m-1}}} \left\| \left\| \frac{\mathbf{v}^T \tilde{\mathbf{X}}^T}{\|\boldsymbol{\Xi}^{\frac{1}{2}} \mathbf{u}\|_2} \cdot \boldsymbol{\Xi}^{\frac{1}{2}} \mathbf{u} \right\| \right\|_{\psi_2} = \sup_{\mathbf{v} \in \mathbb{S}^{p-1}} \left\| \left\| \tilde{\mathbf{X}} \mathbf{v} \right\| \right\|_{\psi_2} \le C\tilde{\kappa}
\end{aligned}
$$

where the inequality follows from noting that the vector $\tilde{\mathbf{X}}\mathbf{v}$ has independent elements with $\psi_2$-norm bounded by $\tilde{\kappa}$, and thus $\left\| \|\tilde{\mathbf{X}}\mathbf{v}\| \right\|_{\psi_2} \le C\tilde{\kappa}$ for any $\mathbf{v} \in \mathbb{S}^{p-1}$. Therefore condition (7) also holds with $\kappa = C\tilde{\kappa}$. ∎

## 2 Proofs for Section 3.1

### 2.1 Proof of Lemma 2

**Statement of Lemma 2:** *Suppose the RE condition* (9) *is satisfied by* $\mathbf{X}_1, \ldots, \mathbf{X}_n$ *and* $\boldsymbol{\Sigma}$ *with* $\alpha > 0$ *for the set* $\mathcal{A}(\boldsymbol{\theta}^*) = \mathrm{cone}\left\{ \mathbf{v} \mid \|\boldsymbol{\theta}^* + \mathbf{v}\| \le \|\boldsymbol{\theta}^*\| \right\} \cap \mathbb{S}^{p-1}$. *If* $\gamma_n$ *is admissible, then* $\hat{\boldsymbol{\theta}}$ *in* (11) *satisfies*

$$\left\| \hat{\boldsymbol{\theta}} - \boldsymbol{\theta}^* \right\|_2 \le 2\Psi(\boldsymbol{\theta}^*) \cdot \frac{\gamma_n}{\alpha} \,, \tag{S.18}$$

*in which* $\Psi(\boldsymbol{\theta}^*)$ *is the restricted norm compatibility defined as* $\Psi(\boldsymbol{\theta}^*) = \sup_{\mathbf{v} \in \mathcal{A}(\boldsymbol{\theta}^*)} \frac{\|\mathbf{v}\|}{\|\mathbf{v}\|_2}$.

*Proof:* Since $\hat{\boldsymbol{\theta}}$ is feasible and $\gamma_n$ is selected to be admissible, we have

$$\left\| \frac{1}{n} \sum_{i=1}^n \mathbf{X}_i^T \boldsymbol{\Sigma}^{-1} (\mathbf{X}_i \hat{\boldsymbol{\theta}} - \mathbf{y}_i) \right\|_* \le \gamma_n, \quad \left\| \frac{1}{n} \sum_{i=1}^n \mathbf{X}_i^T \boldsymbol{\Sigma}^{-1} (\mathbf{X}_i \boldsymbol{\theta}^* - \mathbf{y}_i) \right\|_* \le \gamma_n$$

$$\implies \quad \left\| \frac{1}{n} \sum_{i=1}^n \mathbf{X}_i^T \boldsymbol{\Sigma}^{-1} \mathbf{X}_i (\hat{\boldsymbol{\theta}} - \boldsymbol{\theta}^*) \right\|_* \le 2\gamma_n$$

$$\implies \quad \left\langle \hat{\boldsymbol{\theta}} - \boldsymbol{\theta}^*, \frac{1}{n} \sum_{i=1}^n \mathbf{X}_i^T \boldsymbol{\Sigma}^{-1} \mathbf{X}_i (\hat{\boldsymbol{\theta}} - \boldsymbol{\theta}^*) \right\rangle \le \|\hat{\boldsymbol{\theta}} - \boldsymbol{\theta}^*\| \cdot \left\| \frac{1}{n} \sum_{i=1}^n \mathbf{X}_i^T \boldsymbol{\Sigma}^{-1} \mathbf{X}_i (\hat{\boldsymbol{\theta}} - \boldsymbol{\theta}^*) \right\|_*$$

$$\implies \quad (\hat{\boldsymbol{\theta}} - \boldsymbol{\theta}^*)^T \left( \frac{1}{n} \sum_{i=1}^n \mathbf{X}_i^T \boldsymbol{\Sigma}^{-1} \mathbf{X}_i \right) (\hat{\boldsymbol{\theta}} - \boldsymbol{\theta}^*) \le 2\gamma_n \|\hat{\boldsymbol{\theta}} - \boldsymbol{\theta}^*\|$$

As $\|\hat{\boldsymbol{\theta}}\| \le \|\boldsymbol{\theta}^*\|$, we have $\frac{\hat{\boldsymbol{\theta}} - \boldsymbol{\theta}^*}{\|\hat{\boldsymbol{\theta}} - \boldsymbol{\theta}^*\|_2} \in \mathcal{A}(\boldsymbol{\theta}^*)$. By the assumption of RE condition, we further obtain

$$\alpha \|\hat{\boldsymbol{\theta}} - \boldsymbol{\theta}^*\|_2^2 \le (\hat{\boldsymbol{\theta}} - \boldsymbol{\theta}^*)^T \left( \frac{1}{n} \sum_{i=1}^n \mathbf{X}_i^T \boldsymbol{\Sigma}^{-1} \mathbf{X}_i \right) (\hat{\boldsymbol{\theta}} - \boldsymbol{\theta}^*) \le 2\gamma_n \|\hat{\boldsymbol{\theta}} - \boldsymbol{\theta}^*\|$$

$$\implies \quad \|\hat{\boldsymbol{\theta}} - \boldsymbol{\theta}^*\|_2 \le \frac{\|\hat{\boldsymbol{\theta}} - \boldsymbol{\theta}^*\|}{\|\hat{\boldsymbol{\theta}} - \boldsymbol{\theta}^*\|_2} \cdot \frac{2\gamma_n}{\alpha} \le 2\Psi(\boldsymbol{\theta}^*) \cdot \frac{\gamma_n}{\alpha} \,,$$

where we use the definition of restricted norm compatibility. ∎

## 2.2 Proof of Lemma 3

**Statement of Lemma 3:** *Given sub-Gaussian $\mathbf{X} \in \mathbb{R}^{m \times p}$ with its i.i.d. copies $\mathbf{X}_1, \ldots, \mathbf{X}_n$, and covariance $\mathbf{\Sigma} \in \mathbb{R}^{m \times m}$ with eigenvectors $\mathbf{u}_1, \ldots, \mathbf{u}_m$, let $\mathbf{\Gamma} = \mathbb{E}[\mathbf{X}^T \mathbf{\Sigma}^{-1} \mathbf{X}]$ and $\hat{\mathbf{\Gamma}} = \frac{1}{n} \sum_{i=1}^{n} \mathbf{X}_i^T \mathbf{\Sigma}^{-1} \mathbf{X}_i$. Define the set $\mathcal{A}_{\mathbf{\Gamma}_j}$ for $\mathcal{A} \subseteq \mathbb{S}^{p-1}$ and each $\mathbf{\Gamma}_j = \mathbb{E}[\mathbf{X}^T \mathbf{u}_j \mathbf{u}_j^T \mathbf{X}]$ as $\mathcal{A}_{\mathbf{\Gamma}_j} = \left\{ \mathbf{v} \in \mathbb{S}^{p-1} \mid \mathbf{\Gamma}_j^{-\frac{1}{2}} \mathbf{v} \in \mathrm{cone}(\mathcal{A}) \right\}$. If $n \geq C_1 \kappa^4 \cdot \max_j \left\{ w^2(\mathcal{A}_{\mathbf{\Gamma}_j}) \right\}$, with probability at least $1 - m \exp(-C_2 n / \kappa^4)$, we have*

$$\mathbf{v}^T \hat{\mathbf{\Gamma}} \mathbf{v} \geq \frac{1}{2} \mathbf{v}^T \mathbf{\Gamma} \mathbf{v}, \quad \forall \mathbf{v} \in \mathcal{A} . \tag{S.19}$$

*Proof:* Assume that the eigenvalue decomposition of $\mathbf{\Sigma}$ is given by $\mathbf{\Sigma} = \sum_{i=j}^{m} \sigma_i \mathbf{u}_j \mathbf{u}_j^T$. For convenience, we denote $\mathbf{z}^j = \mathbf{X}^T \mathbf{u}_j$, $\mathbf{z}_i^j = \mathbf{X}_i^T \mathbf{u}_j$, and $\hat{\mathbf{\Gamma}}_j = \frac{1}{n} \sum_{i=1}^{n} \mathbf{X}_i^T \mathbf{u}_j \mathbf{u}_j^T \mathbf{X}_i$. Note that $\mathbf{\Gamma}_j = \mathbb{E}[\mathbf{z}^j \mathbf{z}^{j^T}]$, $\mathbf{\Gamma} = \sum_{i=j}^{m} \frac{\mathbf{\Gamma}_j}{\sigma_j}$, $\hat{\mathbf{\Gamma}}_j = \frac{1}{n} \sum_{i=1}^{n} \mathbf{z}_i^j \mathbf{z}_i^{j^T}$, and $\hat{\mathbf{\Gamma}} = \sum_{j=1}^{m} \frac{\hat{\mathbf{\Gamma}}_j}{\sigma_j}$. In order to apply Lemma C, we let $(\Omega_j, \mu_j)$ be the probability measure that $\mathbf{z}^j$ is defined on, and construct the function set

$$\mathcal{H}_j = \left\{ h_{\mathbf{v}} = \left\langle \mathbf{\Gamma}_j^{-\frac{1}{2}} \mathbf{v}, \cdot \right\rangle \mid \mathbf{v} \in \mathcal{A}_{\mathbf{\Gamma}_j} \right\}$$

It is easy to see that for any $h_{\mathbf{v}} \in \mathcal{H}_j$,

$$\mathbb{E}[h_{\mathbf{v}}^2] = \mathbb{E}_{\mathbf{z}^j \sim \mu_j} \left[ \mathbf{v}^T \mathbf{\Gamma}_j^{-\frac{1}{2}} \mathbf{z}^j \mathbf{z}^{j^T} \mathbf{\Gamma}_j^{-\frac{1}{2}} \mathbf{v} \right] = \mathbf{v}^T \mathbf{\Gamma}_j^{-\frac{1}{2}} \left( \mathbb{E}_{\mathbf{z}^j \sim \mu_j} \left[ \mathbf{z}^j \mathbf{z}^{j^T} \right] \right) \mathbf{\Gamma}_j^{-\frac{1}{2}} \mathbf{v} = \mathbf{v}^T \mathbf{v} = 1 ,$$

i.e., $\mathcal{H}_j \subseteq \mathbb{S}_{L_2(\mu_j)} = \{ h \mid \|h\|_{L_2(\mu_j)} = 1 \}$. Based on the definition of sub-Gaussian $\mathbf{X}$, we also have for any $\mathbf{v} \in \mathcal{A}_{\mathbf{\Gamma}_j}$,

$$\|h_{\mathbf{v}}\|_{\psi_2} = \left\| \left\langle \mathbf{\Gamma}_j^{-\frac{1}{2}} \mathbf{v}, \mathbf{z}^j \right\rangle \right\|_{\psi_2} = \left\| \mathbf{v}^T \mathbf{\Gamma}_j^{-\frac{1}{2}} \mathbf{X}^T \mathbf{u}_j \right\|_{\psi_2} \leq \kappa ,$$

and also for any $\mathbf{v}_1, \mathbf{v}_2 \in \mathcal{A}_{\mathbf{\Gamma}_j}$, we have

$$\|h_{\mathbf{v}_1} - h_{\mathbf{v}_2}\|_{\psi_2} = \left\| (\mathbf{v}_1 - \mathbf{v}_2)^T \mathbf{\Gamma}_j^{-\frac{1}{2}} \mathbf{z}^j \right\|_{\psi_2} \leq \kappa \cdot \|\mathbf{v}_1 - \mathbf{v}_2\|_2 .$$

If we choose $\beta = \frac{1}{2}$, using (S.7), (S.8) and (S.9), then we have

$$c_1 \kappa \cdot \gamma_2(\mathcal{H}_j, \|\cdot\|_{\psi_2}) \leq c_1 \kappa^2 \cdot \gamma_2(\mathcal{A}_{\mathbf{\Gamma}_j}, \|\cdot\|_2) \leq c_1 c_4 \kappa^2 \cdot w(\mathcal{A}_{\mathbf{\Gamma}_j}) \leq \beta \sqrt{n}$$

when $n \geq C_1 \kappa^4 w^2(\mathcal{A}_{\mathbf{\Gamma}_j})$ where $C_1 = 4 c_1^2 c_4^2$. By Lemma C, with probability at least $1 - \exp(-c_2 \beta^2 n / \kappa^4) = 1 - \exp(-C_2 n / \kappa^4)$ where $C_2 = c_2 / 4$, we have

$$\sup_{h \in \mathcal{H}_j} \left| \frac{1}{n} \sum_{i=1}^{n} h^2(\mathbf{z}_i^j) - \mathbb{E}[h^2] \right| = \sup_{\mathbf{v} \in \mathcal{A}_{\mathbf{\Gamma}_j}} \left| \frac{1}{n} \sum_{i=1}^{n} \mathbf{v}^T \mathbf{\Gamma}_j^{-\frac{1}{2}} \mathbf{z}_i^j \mathbf{z}_i^{j^T} \mathbf{\Gamma}_j^{-\frac{1}{2}} \mathbf{v} - 1 \right|$$

$$= \sup_{\mathbf{v} \in \mathcal{A}_{\mathbf{\Gamma}_j}} \left| \mathbf{v}^T \mathbf{\Gamma}_j^{-\frac{1}{2}} \hat{\mathbf{\Gamma}}_j \mathbf{\Gamma}_j^{-\frac{1}{2}} \mathbf{v} - 1 \right| \leq \frac{1}{2}$$

$$\implies \mathbf{v}^T \mathbf{\Gamma}_j^{-\frac{1}{2}} \hat{\mathbf{\Gamma}}_j \mathbf{\Gamma}_j^{-\frac{1}{2}} \mathbf{v} \geq \frac{1}{2}, \quad \forall \mathbf{v} \in \mathcal{A}_{\mathbf{\Gamma}_j}$$

$$\implies \mathbf{v}^T \mathbf{\Gamma}_j^{-\frac{1}{2}} \hat{\mathbf{\Gamma}}_j \mathbf{\Gamma}_j^{-\frac{1}{2}} \mathbf{v} \geq \frac{1}{2} \left( \mathbf{v}^T \mathbf{\Gamma}_j^{-\frac{1}{2}} \mathbf{\Gamma}_j \mathbf{\Gamma}_j^{-\frac{1}{2}} \mathbf{v} \right), \quad \forall \mathbf{v} \in \mathcal{A}_{\mathbf{\Gamma}_j}$$

Let $\mathbf{w} = \mathbf{\Gamma}_j^{-\frac{1}{2}} \mathbf{v}$, and note that the inequalities above are preserved under arbitrary scaling of $\mathbf{w}$. By recalling the definition of $\mathcal{A}_{\mathbf{\Gamma}_j}$, it is not difficult to see that

$$\mathbf{w}^T \hat{\mathbf{\Gamma}}_j \mathbf{w} \geq \frac{1}{2} \mathbf{w}^T \mathbf{\Gamma}_j \mathbf{w}, \quad \forall \mathbf{w} \in \mathcal{A} . \tag{S.20}$$

Combining (S.20) for each $\mathbf{\Gamma}_j$ using union bound, we obtain

$$\mathbf{w}^T \left( \sum_{i=1}^{m} \frac{\hat{\mathbf{\Gamma}}_j}{\sigma_j} \right) \mathbf{w} \geq \frac{1}{2} \mathbf{w}^T \left( \sum_{i=1}^{m} \frac{\mathbf{\Gamma}_j}{\sigma_j} \right) \mathbf{w}, \ \forall \mathbf{w} \in \mathcal{A} \implies \mathbf{w}^T \hat{\mathbf{\Gamma}} \mathbf{w} \geq \frac{1}{2} \mathbf{w}^T \mathbf{\Gamma} \mathbf{w}, \ \forall \mathbf{w} \in \mathcal{A} ,$$

which completes the proof by renaming $\mathbf{w}$ as $\mathbf{v}$. ∎

## 2.3 Proof of Lemma 4

**Statement of Lemma 4:** *Let $\kappa_0$ be the $\psi_2$-norm of standard Gaussian random vector and $\Gamma_{\mathbf{u}} = \mathbb{E}[\mathbf{X}^T\mathbf{u}\mathbf{u}^T\mathbf{X}]$, where $\mathbf{u} \in \mathbb{S}^{m-1}$ is fixed. For $\mathcal{A}_{\Gamma_{\mathbf{u}}}$ defined in Lemma 3, we have*

$$w(\mathcal{A}_{\Gamma_{\mathbf{u}}}) \leq C\kappa_0\sqrt{\mu_{\max}/\mu_{\min}} \cdot (w(\mathcal{A}) + 3) \ , \tag{S.21}$$

*Proof:* Recall the definition of Gaussian width $w(\mathcal{A}_{\Gamma_{\mathbf{u}}}) = \mathbb{E}\left[\sup_{\mathbf{v} \in \mathcal{A}_{\Gamma_{\mathbf{u}}}} \langle \mathbf{v}, \mathbf{g}\rangle\right]$, where $\mathbf{g}$ is a standard Gaussian random vector. Given the assumption (8), we have $\mu_{\min} \leq \lambda_{\min}(\Gamma_{\mathbf{u}}) \leq \lambda_{\max}(\Gamma_{\mathbf{u}}) \leq \mu_{\max}$, and note that

$$\sup_{\mathbf{v} \in \mathcal{A}_{\Gamma_{\mathbf{u}}}} \langle \mathbf{v}, \mathbf{g}\rangle = \sup_{\mathbf{v} \in \mathcal{A}_{\Gamma_{\mathbf{u}}}} \left\langle \Gamma_{\mathbf{u}}^{-\frac{1}{2}}\mathbf{v}, \Gamma_{\mathbf{u}}^{\frac{1}{2}}\mathbf{g}\right\rangle \leq \sup_{\mathbf{v} \in \mathrm{cone}(\mathcal{A}) \cap \frac{1}{\sqrt{\mu_{\min}}}\mathbb{B}^p} \left\langle \mathbf{v}, \Gamma_{\mathbf{u}}^{\frac{1}{2}}\mathbf{g}\right\rangle$$
$$= \frac{1}{\sqrt{\mu_{\min}}} \cdot \sup_{\mathbf{v} \in \mathrm{cone}(\mathcal{A}) \cap \mathbb{B}^p} \left\langle \mathbf{v}, \Gamma_{\mathbf{u}}^{\frac{1}{2}}\mathbf{g}\right\rangle \ , \tag{S.22}$$

where the inequality follows from $\Gamma_{\mathbf{u}}^{-\frac{1}{2}}\mathbf{v} \in \mathrm{cone}(\mathcal{A})$ and $\|\Gamma_{\mathbf{u}}^{-\frac{1}{2}}\mathbf{v}\|_2 \leq \frac{1}{\sqrt{\mu_{\min}}}$. Now we use generic chaining to bound the right-hand side above. Denote the set $\mathrm{cone}(\mathcal{A}) \cap \mathbb{B}^p$ by $\mathcal{T}$, and we consider the stochastic process $\{Z_{\mathbf{v}} = \langle \mathbf{v}, \Gamma_{\mathbf{u}}^{\frac{1}{2}}\mathbf{g}\rangle\}_{\mathbf{v} \in \mathcal{T}}$. For any $\mathbf{v}_1, \mathbf{v}_2 \in \mathcal{T}$, we have

$$\|Z_{\mathbf{v}_1} - Z_{\mathbf{v}_2}\|_{\psi_2} = \left\|\langle \Gamma_{\mathbf{u}}^{\frac{1}{2}}(\mathbf{v}_1 - \mathbf{v}_2), \mathbf{g}\rangle\right\|_{\psi_2} \leq \kappa_0 \left\|\Gamma_{\mathbf{u}}^{\frac{1}{2}}(\mathbf{v}_1 - \mathbf{v}_2)\right\|_2 \leq \kappa_0\sqrt{\mu_{\max}} \cdot \|\mathbf{v}_1 - \mathbf{v}_2\|_2 \ .$$

If we define for $\mathcal{T}$ the metric $s(\mathbf{v}_1, \mathbf{v}_2) = \kappa_0\sqrt{\mu_{\max}} \cdot \|\mathbf{v}_1 - \mathbf{v}_2\|_2$, it follows from Proposition A that

$$\mathbb{P}\left(|Z_{\mathbf{v}_1} - Z_{\mathbf{v}_2}| \geq \epsilon\right) \leq e \cdot \exp\left(-\frac{c\epsilon^2}{\kappa_0^2\mu_{\max}\|\mathbf{v}_1 - \mathbf{v}_2\|_2^2}\right) = e \cdot \exp\left(-\frac{c\epsilon^2}{s^2(\mathbf{v}_1, \mathbf{v}_2)}\right) \ .$$

By Lemma B, (S.8) and (S.16), we obtain

$$\mathbb{E}\left[\sup_{\mathbf{v} \in \mathcal{T}}\langle \mathbf{v}, \Gamma_{\mathbf{u}}^{\frac{1}{2}}\mathbf{g}\rangle\right] = \mathbb{E}\left[\sup_{\mathbf{v} \in \mathcal{T}} Z_{\mathbf{v}}\right] \leq c_1\gamma_2(\mathcal{T}, s) = c_1\kappa_0\sqrt{\mu_{\max}}\gamma_2(\mathcal{T}, \|\cdot\|_2) \leq c_1 c_2\kappa_0\sqrt{\mu_{\max}} \cdot w(\mathcal{T}) \tag{S.23}$$

Note that $\mathcal{T} = \mathrm{cone}(\mathcal{A}) \cap \mathbb{B}^p \subseteq \mathrm{conv}(\mathcal{A} \cup \{\mathbf{0}\})$. By Lemma E, we have

$$w(\mathcal{T}) \leq w(\mathrm{conv}(\mathcal{A} \cup \{\mathbf{0}\})) = w(\mathcal{A} \cup \{\mathbf{0}\}) \leq \max\{w(\mathcal{A}), w(\mathbf{0})\} + 2\sqrt{\ln 4} \leq w(\mathcal{A}) + 3 \ . \tag{S.24}$$

Combining (S.22), (S.23) and (S.24), we have

$$w(\mathcal{A}_{\Gamma_{\mathbf{u}}}) = \mathbb{E}\left[\sup_{\mathbf{v} \in \mathcal{A}_{\Gamma_{\mathbf{u}}}} \langle \mathbf{v}, \mathbf{g}\rangle\right] \leq \frac{1}{\sqrt{\mu_{\min}}}\mathbb{E}\left[\sup_{\mathbf{v} \in \mathcal{T}} \left\langle \mathbf{v}, \Gamma_{\mathbf{u}}^{\frac{1}{2}}\mathbf{g}\right\rangle\right] \leq c_1 c_2\kappa_0\sqrt{\frac{\mu_{\max}}{\mu_{\min}}} \cdot (w(\mathcal{A}) + 3) \ , \tag{S.25}$$

where the last inequality follows from condition (8). ∎

## 2.4 Proof of Corollary 1

**Statement of Corollary 1:** *Under the notations of Lemma 3 and 4, if $n \geq C_1\kappa_0^2\kappa^4 \cdot \frac{\mu_{\max}}{\mu_{\min}} \cdot (w(\mathcal{A}) + 3)^2$, then the following inequality holds for all $\mathbf{v} \in \mathcal{A} \subseteq \mathbb{S}^{p-1}$ with probability at least $1 - m\exp(-C_2 n/\kappa^4)$,*

$$\mathbf{v}^T\hat{\Gamma}\mathbf{v} \geq \frac{\mu_{\min}}{2} \cdot \mathrm{Tr}(\Sigma^{-1}) \tag{S.26}$$

*Proof:* Given the definition of sub-Gaussian $\mathbf{X}$ and Lemma 3, we have

$$\mathbf{v}^T\hat{\Gamma}\mathbf{v} \geq \frac{1}{2}\mathbf{v}^T\Gamma\mathbf{v} = \frac{1}{2}\mathbf{v}^T\left(\sum_{j=1}^m \frac{1}{\sigma_j} \cdot \mathbb{E}\left[\mathbf{X}^T\mathbf{u}_j\mathbf{u}_j^T\mathbf{X}\right]\right)\mathbf{v}$$

$$\geq \frac{\mu_{\min}}{2} \cdot \mathbf{v}^T\mathbf{v}\left(\sum_{j=1}^m \frac{1}{\sigma_j}\right) = \frac{\mu_{\min}}{2}\mathrm{Tr}\left(\Sigma^{-1}\right) \ .$$

Using the bound in Lemma 4, we have

$$n \geq C_1 \kappa_0^2 \kappa^4 \cdot \frac{\mu_{\max}}{\mu_{\min}} \cdot (w(\mathcal{A}) + 3)^2 \quad \Longrightarrow \quad n \geq C\kappa^4 \cdot \max_j \left\{ w^2(\mathcal{A}_{\mathbf{\Gamma}_j}) \right\}$$

We complete the proof by combining the two equations above. ∎

## 2.5 Proof of Lemma 5

**Statement of Lemma 5:** *Assume that $\mathbf{X}_i$ is sub-Gaussian and $\boldsymbol{\eta}_i \sim \mathcal{N}(\mathbf{0}, \boldsymbol{\Sigma}_*)$. The following inequality holds with probability at least $1 - \exp\left(-\frac{n\tau^2}{2}\right) - C_2 \exp\left(-\frac{C_1^2 w^2(\mathcal{B})}{4\rho^2}\right)$*

$$\left\| \frac{1}{n} \sum_{i=1}^n \mathbf{X}_i^T \boldsymbol{\Sigma}^{-1} \boldsymbol{\eta}_i \right\|_* \leq \frac{C\kappa\sqrt{\mu_{\max}}}{\sqrt{n}} \cdot \sqrt{\mathrm{Tr}\left(\boldsymbol{\Sigma}^{-1} \boldsymbol{\Sigma}_* \boldsymbol{\Sigma}^{-1}\right)} \cdot w(\mathcal{B}) , \qquad \text{(S.27)}$$

*where $\mathcal{B}$ denotes the unit ball of norm $\| \cdot \|$, $\rho = \sup_{\mathbf{v} \in \mathcal{B}} \|\mathbf{v}\|_2$, and $\tau = \|\boldsymbol{\Sigma}^{-1} \boldsymbol{\Sigma}_*^{\frac{1}{2}}\|_F / \|\boldsymbol{\Sigma}^{-1} \boldsymbol{\Sigma}_*^{\frac{1}{2}}\|_2$.*

*Proof:* Since design $\mathbf{X}_i$ and noise $\boldsymbol{\eta}_i$ are independent, we first consider the scenario where each $\boldsymbol{\eta}_i$ is arbitrary but fixed vector. Using the definition of dual norm, we have

$$\left\| \frac{1}{n} \sum_{i=1}^n \mathbf{X}_i^T \boldsymbol{\Sigma}^{-1} \boldsymbol{\eta}_i \right\|_* = \frac{1}{n} \cdot \sup_{\mathbf{v} \in \mathcal{B}} \left\langle \mathbf{v}, \sum_{i=1}^n \mathbf{X}_i^T \boldsymbol{\Sigma}^{-1} \boldsymbol{\eta}_i \right\rangle = \frac{1}{n} \cdot \sup_{\mathbf{v} \in \mathcal{B}} \sum_{i=1}^n \left\langle \boldsymbol{\Lambda}_i^{\frac{1}{2}} \mathbf{v}, \boldsymbol{\Lambda}_i^{-\frac{1}{2}} \mathbf{X}_i^T \boldsymbol{\Sigma}^{-1} \boldsymbol{\eta}_i \right\rangle$$

where $\boldsymbol{\Lambda}_i = \mathbb{E}_{\mathbf{X}_i}[\mathbf{X}_i^T \boldsymbol{\Sigma}^{-1} \boldsymbol{\eta}_i \boldsymbol{\eta}_i^T \boldsymbol{\Sigma}^{-1} \mathbf{X}_i]$. Based on the definition of sub-Gaussian $\mathbf{X}_i$, we get

$$\left\| \boldsymbol{\Lambda}_i^{-\frac{1}{2}} \mathbf{X}_i^T \boldsymbol{\Sigma}^{-1} \boldsymbol{\eta}_i \right\|_{\psi_2} \leq \kappa \quad \Longrightarrow$$

$$\left\| \sum_{i=1}^n \left\langle \boldsymbol{\Lambda}_i^{\frac{1}{2}} \mathbf{v}, \boldsymbol{\Lambda}_i^{-\frac{1}{2}} \mathbf{X}_i^T \boldsymbol{\Sigma}^{-1} \boldsymbol{\eta}_i \right\rangle \right\|_{\psi_2} \leq c_0 \max_{1 \leq i \leq n} \left\| \boldsymbol{\Lambda}_i^{-\frac{1}{2}} \mathbf{X}_i^T \boldsymbol{\Sigma}^{-1} \boldsymbol{\eta}_i \right\|_{\psi_2} \cdot \sqrt{\sum_{i=1}^n \left\| \boldsymbol{\Lambda}_i^{\frac{1}{2}} \mathbf{v} \right\|_2^2}$$

$$\leq c_0 \kappa \sqrt{\sum_{i=1}^n \left\| \boldsymbol{\Lambda}_i^{\frac{1}{2}} \right\|_2^2 \|\mathbf{v}\|_2^2} \leq c_0 \kappa \sqrt{\mu_{\max}} \cdot \sqrt{\sum_{i=1}^n \|\boldsymbol{\Sigma}^{-1} \boldsymbol{\eta}_i\|_2^2} \cdot \|\mathbf{v}\|_2$$

where we use Lemma A in the first inequality by treating the sum of inner products as one "big" inner product. The last inequality follows from the definition of $\mu_{\max}$ in (8). Now we consider the stochastic process $\left\{ Z_{\mathbf{v}} = \left\langle \mathbf{v}, \sum_{i=1}^n \mathbf{X}_i^T \boldsymbol{\Sigma}^{-1} \boldsymbol{\eta}_i \right\rangle \right\}_{\mathbf{v} \in \mathcal{B}}$, where $\boldsymbol{\eta}_i$ is still fixed. For any $Z_{\mathbf{v}_1}$ and $Z_{\mathbf{v}_2}$, by the argument above and Proposition A, we have

$$\|Z_{\mathbf{v}_1} - Z_{\mathbf{v}_2}\|_{\psi_2} \leq c_0 \kappa \sqrt{\mu_{\max}} \cdot \sqrt{\sum_{i=1}^n \|\boldsymbol{\Sigma}^{-1} \boldsymbol{\eta}_i\|_2^2} \cdot \|\mathbf{v}_1 - \mathbf{v}_2\|_2 \triangleq s(\mathbf{v}_1, \mathbf{v}_2)$$

$$\Longrightarrow \quad \mathbb{P}\left(|Z_{\mathbf{v}_1} - Z_{\mathbf{v}_2}| > \epsilon\right) \leq e \cdot \exp\left(-\frac{C_1 \epsilon^2}{s^2(\mathbf{v}_1, \mathbf{v}_2)}\right)$$

It follows from (S.8), (S.16) and Lemma B that

$$\gamma_2(\mathcal{B}, s) = c_0 \kappa \sqrt{\mu_{\max}} \cdot \sqrt{\sum_{i=1}^n \|\boldsymbol{\Sigma}^{-1} \boldsymbol{\eta}_i\|_2^2} \cdot \gamma_2(\mathcal{B}, \| \cdot \|_2) \leq c_0 c_1 \kappa \sqrt{\mu_{\max}} \cdot \sqrt{\sum_{i=1}^n \|\boldsymbol{\Sigma}^{-1} \boldsymbol{\eta}_i\|_2^2} \cdot w(\mathcal{B}) ,$$

$$\mathbb{P}_{\mathbf{X}_i}\left( \sup_{\mathbf{v}_1, \mathbf{v}_2 \in \mathcal{B}} |Z_{\mathbf{v}_1} - Z_{\mathbf{v}_2}| \geq c_2 \left(\gamma_2(\mathcal{B}, s) + \epsilon \cdot \mathrm{diam}\left(\mathcal{B}, s\right)\right) \right) \leq c_3 \exp\left(-\epsilon^2\right)$$

Combining the two inequalities above with the symmetry of $\mathcal{B}$, we obtain

$$\mathbb{P}_{\mathbf{X}}\left( \sup_{\mathbf{v} \in \mathcal{B}} Z_{\mathbf{v}} \geq c_0 c_2 \kappa \sqrt{\mu_{\max}} \cdot \sqrt{\sum_{i=1}^n \|\boldsymbol{\Sigma}^{-1} \boldsymbol{\eta}_i\|_2^2} \left(\frac{c_1}{2} \cdot w(\mathcal{B}) + \epsilon \cdot \sup_{\mathbf{v} \in \mathcal{B}} \|\mathbf{v}\|_2\right) \right) \leq c_3 \exp\left(-\epsilon^2\right)$$

Letting $\rho = \sup_{\mathbf{v} \in \mathcal{B}} \|\mathbf{v}\|_2$, $\epsilon = \frac{c_1 w(\mathcal{B})}{2\rho}$, with probability at least $1 - c_3 \exp(-\frac{c_1^2 w^2(\mathcal{B})}{4\rho^2})$, we have

$$\sup_{\mathbf{v} \in \mathcal{B}} Z_{\mathbf{v}} = \left\| \sum_{i=1}^n \mathbf{X}_i^T \mathbf{\Sigma}^{-1} \boldsymbol{\eta}_i \right\|_* \leq c_0 c_1 c_2 \kappa \sqrt{\mu_{\max}} \cdot \sqrt{\sum_{i=1}^n \|\mathbf{\Sigma}^{-1} \boldsymbol{\eta}_i\|_2^2} \cdot w(\mathcal{B}) \tag{S.28}$$

for any given set of $\boldsymbol{\eta}_i$. Now we incorporate the randomness of $\boldsymbol{\eta}_i$. Essentially we need to bound

$$\sqrt{\sum_{i=1}^n \|\mathbf{\Sigma}^{-1} \boldsymbol{\eta}_i\|_2^2} = \sqrt{\sum_{i=1}^n \left\| \mathbf{\Sigma}^{-1} \mathbf{\Sigma}_*^{\frac{1}{2}} \tilde{\boldsymbol{\eta}}_i \right\|_2^2},$$

where each $\tilde{\boldsymbol{\eta}}_i$ is an $m$-dimensional standard (isotropic) Gaussian random vector. Given $\mathbf{v} = [\mathbf{v}_1^T, \ldots, \mathbf{v}_n^T]^T \in \mathbb{R}^{mn}$, Denote $f(\mathbf{v}) = \sqrt{\sum_{i=1}^n \left\| \mathbf{\Sigma}^{-1} \mathbf{\Sigma}_*^{\frac{1}{2}} \mathbf{v}_i \right\|_2^2}$, and we have

$$|f(\mathbf{v}) - f(\mathbf{w})| = \left| \sqrt{\sum_{i=1}^n \left\| \mathbf{\Sigma}^{-1} \mathbf{\Sigma}_*^{\frac{1}{2}} \mathbf{v}_i \right\|_2^2} - \sqrt{\sum_{i=1}^n \left\| \mathbf{\Sigma}^{-1} \mathbf{\Sigma}_*^{\frac{1}{2}} \mathbf{w}_i \right\|_2^2} \right|$$

$$\leq \sqrt{\sum_{i=1}^n \left( \left\| \mathbf{\Sigma}^{-1} \mathbf{\Sigma}_*^{\frac{1}{2}} \mathbf{v}_i \right\|_2 - \left\| \mathbf{\Sigma}^{-1} \mathbf{\Sigma}_*^{\frac{1}{2}} \mathbf{w}_i \right\|_2 \right)^2}$$

$$\leq \sqrt{\sum_{i=1}^n \left\| \mathbf{\Sigma}^{-1} \mathbf{\Sigma}_*^{\frac{1}{2}} (\mathbf{v}_i - \mathbf{w}_i) \right\|_2^2}$$

$$\leq \sqrt{\sum_{i=1}^n \left\| \mathbf{\Sigma}^{-1} \mathbf{\Sigma}_*^{\frac{1}{2}} \right\|_2^2 \|\mathbf{v}_i - \mathbf{w}_i\|_2^2} = \left\| \mathbf{\Sigma}^{-1} \mathbf{\Sigma}_*^{\frac{1}{2}} \right\|_2 \|\mathbf{v} - \mathbf{w}\|_2$$

which implies that $f$ is a Lipschitz function with parameter $\|\mathbf{\Sigma}^{-1} \mathbf{\Sigma}_*^{\frac{1}{2}}\|_2$. The first two inequalities use the triangular inequality for $L_2$ norm. Letting $\tilde{\boldsymbol{\eta}} = [\tilde{\boldsymbol{\eta}}_1^T, \ldots, \tilde{\boldsymbol{\eta}}_n^T]^T$, by the concentration inequality for Lipschitz function of Gaussian random vector (see Proposition 5.34 in [6]), we obtain

$$\mathbb{P}\left( f(\tilde{\boldsymbol{\eta}}) - \mathbb{E}f(\tilde{\boldsymbol{\eta}}) > t \right) \leq \exp\left( \frac{-t^2}{2\|\mathbf{\Sigma}^{-1} \mathbf{\Sigma}_*^{\frac{1}{2}}\|_2^2} \right)$$

$$\implies \mathbb{P}\left( \sqrt{\sum_{i=1}^n \left\| \mathbf{\Sigma}^{-1} \mathbf{\Sigma}_*^{\frac{1}{2}} \tilde{\boldsymbol{\eta}}_i \right\|_2^2} - \mathbb{E}\sqrt{\sum_{i=1}^n \left\| \mathbf{\Sigma}^{-1} \mathbf{\Sigma}_*^{\frac{1}{2}} \tilde{\boldsymbol{\eta}}_i \right\|_2^2} > t \right) \leq \exp\left( \frac{-t^2}{2\|\mathbf{\Sigma}^{-1} \mathbf{\Sigma}_*^{\frac{1}{2}}\|_2^2} \right)$$

$$\implies \mathbb{P}\left( \sqrt{\sum_{i=1}^n \|\mathbf{\Sigma}^{-1} \boldsymbol{\eta}_i\|_2^2} - \sqrt{\mathbb{E}\sum_{i=1}^n \mathrm{Tr}\left( \mathbf{\Sigma}^{-1} \mathbf{\Sigma}_*^{\frac{1}{2}} \tilde{\boldsymbol{\eta}}_i \tilde{\boldsymbol{\eta}}_i^T \mathbf{\Sigma}_*^{\frac{1}{2}} \mathbf{\Sigma}^{-1} \right)} > t \right) \leq \exp\left( \frac{-t^2}{2\|\mathbf{\Sigma}^{-1} \mathbf{\Sigma}_*^{\frac{1}{2}}\|_2^2} \right)$$

$$\implies \mathbb{P}\left( \sqrt{\sum_{i=1}^n \|\mathbf{\Sigma}^{-1} \boldsymbol{\eta}_i\|_2^2} - \sqrt{n}\sqrt{\mathrm{Tr}\left( \mathbf{\Sigma}^{-1} \mathbf{\Sigma}_* \mathbf{\Sigma}^{-1} \right)} > t \right) \leq \exp\left( \frac{-t^2}{2\|\mathbf{\Sigma}^{-1} \mathbf{\Sigma}_*^{\frac{1}{2}}\|_2^2} \right)$$

where we use Jensen's inequality in the third step for bounding the expectation $\mathbb{E}f(\tilde{\boldsymbol{\eta}})$. Letting $t = \sqrt{\mathrm{Tr}\left( \mathbf{\Sigma}^{-1} \mathbf{\Sigma}_* \mathbf{\Sigma}^{-1} \right) \cdot n}$ and $\tau = \|\mathbf{\Sigma}^{-1} \mathbf{\Sigma}_*^{\frac{1}{2}}\|_F / \|\mathbf{\Sigma}^{-1} \mathbf{\Sigma}_*^{\frac{1}{2}}\|_2$, with probability at least $1 - \exp\left( -\frac{n\tau^2}{2} \right)$, we have

$$\sqrt{\sum_{i=1}^n \|\mathbf{\Sigma}^{-1} \boldsymbol{\eta}_i\|_2^2} \leq 2\sqrt{n} \cdot \sqrt{\mathrm{Tr}\left( \mathbf{\Sigma}^{-1} \mathbf{\Sigma}_* \mathbf{\Sigma}^{-1} \right)}, \tag{S.29}$$

where we use the relation $\mathrm{Tr}\left( \mathbf{\Sigma}^{-1} \mathbf{\Sigma}_* \mathbf{\Sigma}^{-1} \right) = \|\mathbf{\Sigma}^{-1} \mathbf{\Sigma}_*^{\frac{1}{2}}\|_F^2$. By applying a union bound to (S.28) and (S.29), with probability at least $1 - \exp\left( -\frac{n\tau^2}{2} \right) - c_3 \exp(-\frac{c_1^2 w^2(\mathcal{B})}{4\rho^2})$, the following inequality

holds
$$\left\| \frac{1}{n} \sum_{i=1}^{n} \mathbf{X}_i^T \mathbf{\Sigma}^{-1} \boldsymbol{\eta}_i \right\|_* \leq \frac{2c_0 c_1 c_2 \cdot \kappa \sqrt{\mu_{\max}}}{\sqrt{n}} \cdot \sqrt{\operatorname{Tr}\left(\mathbf{\Sigma}^{-1}\mathbf{\Sigma}_*\mathbf{\Sigma}^{-1}\right)} \cdot w(\mathcal{B}) \qquad \text{(S.30)}$$
Finally we complete the proof by letting $C = 2c_0 c_1 c_2$, $C_1 = c_1$, and $C_2 = c_3$. ∎

## 2.6 Proof of Theorem 1

**Statement of Theorem 1:** *Under the setting of Lemma 5, if $n \geq C_1 \kappa_0^2 \kappa^4 \cdot \frac{\mu_{\max}}{\mu_{\min}} \cdot (w(\mathcal{A}(\boldsymbol{\theta}^*)) + 3)^2$, and $\gamma_n$ is set to $C_2 \kappa \sqrt{\frac{\mu_{\max} \operatorname{Tr}(\mathbf{\Sigma}^{-1}\mathbf{\Sigma}_*\mathbf{\Sigma}^{-1})}{n}} \cdot w(\mathcal{B})$, the estimation error of $\hat{\boldsymbol{\theta}}$ given by (11) satisfies*

$$\|\hat{\boldsymbol{\theta}} - \boldsymbol{\theta}^*\|_2 \leq C\kappa \sqrt{\frac{\mu_{\max}}{\mu_{\min}^2}} \cdot \frac{\sqrt{\operatorname{Tr}\left(\mathbf{\Sigma}^{-1}\mathbf{\Sigma}_*\mathbf{\Sigma}^{-1}\right)}}{\operatorname{Tr}\left(\mathbf{\Sigma}^{-1}\right)} \cdot \frac{\Psi(\boldsymbol{\theta}^*) \cdot w(\mathcal{B})}{\sqrt{n}} , \qquad \text{(S.31)}$$

*with probability at least $1 - m \exp\left(-\frac{C_3 n}{\kappa^4}\right) - \exp\left(-\frac{n\tau^2}{2}\right) - C_4 \exp\left(-\frac{C_5^2 w^2(\mathcal{B})}{4\rho^2}\right)$.*

*Proof:* By Corollary 1, we have the RE condition hold with $\alpha = \frac{\mu_{\min}}{2} \cdot \operatorname{Tr}(\mathbf{\Sigma}^{-1})$ for $\mathcal{A}(\boldsymbol{\theta}^*)$. Combining Lemma 2 and 5, we get

$$\|\hat{\boldsymbol{\theta}} - \boldsymbol{\theta}^*\|_2 \leq 2\Psi(\boldsymbol{\theta}^*) \cdot \frac{\gamma_n}{\alpha} \leq C\kappa \sqrt{\frac{\mu_{\max}}{\mu_{\min}^2}} \cdot \frac{\sqrt{\operatorname{Tr}\left(\mathbf{\Sigma}^{-1}\mathbf{\Sigma}_*\mathbf{\Sigma}^{-1}\right)}}{\operatorname{Tr}\left(\mathbf{\Sigma}^{-1}\right)} \cdot \frac{\Psi(\boldsymbol{\theta}^*) \cdot w(\mathcal{B})}{\sqrt{n}} , \qquad \text{(S.32)}$$

and the probability is computed via union bound. ∎

## 2.7 Proof of Lemma 6

**Statement of Lemma 6:** *Assume $\mathbf{X}$ is defined as in Lemma 1 such that $\mathbf{X} = \mathbf{\Xi}^{\frac{1}{2}} \tilde{\mathbf{X}} \mathbf{\Lambda}^{\frac{1}{2}}$, and rows of $\tilde{\mathbf{X}}$ are i.i.d. with $\|\|\tilde{\mathbf{x}}_j\|\| \leq \tilde{\kappa}$. If $mn \geq C_1 \kappa_0^2 \tilde{\kappa}^4 \cdot \frac{\lambda_{\max}(\mathbf{\Xi})\lambda_{\max}(\mathbf{\Lambda})}{\lambda_{\min}(\mathbf{\Xi})\lambda_{\min}(\mathbf{\Lambda})} \cdot (w(\mathcal{A}) + 3)^2$, with probability at least $1 - \exp(-C_2 mn/\tilde{\kappa}^4)$, the following inequality is satisfied by all $\mathbf{v} \in \mathcal{A} \subseteq \mathbb{S}^{p-1}$,*

$$\mathbf{v}^T \hat{\mathbf{\Gamma}} \mathbf{v} \geq \frac{m}{2} \cdot \lambda_{\min}\left(\mathbf{\Xi}^{\frac{1}{2}} \mathbf{\Sigma}^{-1} \mathbf{\Xi}^{\frac{1}{2}}\right) \cdot \lambda_{\min}(\mathbf{\Lambda}) . \qquad \text{(S.33)}$$

*Proof:* Let $\tilde{\mathbf{x}}_i^{j^T}$ denote the $j$-th row of $\tilde{\mathbf{X}}_i$, which is identically distributed as $\tilde{\mathbf{x}}$. In order to use Lemma C, we let $(\Omega, \mu)$ be the probability measure that $\tilde{\mathbf{x}}$ is defined on. Construct the set of points $\mathcal{A}_{\mathbf{\Lambda}} = \left\{ \mathbf{v} \in \mathbb{S}^{p-1} \mid \mathbf{\Lambda}^{-\frac{1}{2}} \mathbf{v} \in \operatorname{cone}(\mathcal{A}) \right\}$ and the function set

$$\mathcal{H} = \{h_{\mathbf{v}} = \langle \mathbf{v}, \cdot \rangle \mid \mathbf{v} \in \mathcal{A}_{\mathbf{\Lambda}}\}$$

Since $\mathcal{A}_{\mathbf{\Lambda}} \subseteq \mathbb{S}^{p-1}$ and $\tilde{\mathbf{x}}$ is isotropic, it is easy to verify that $\mathbb{E}[h_{\mathbf{v}}^2] = \mathbb{E}_{\tilde{\mathbf{x}} \sim \mu}[\langle \tilde{\mathbf{x}}, \mathbf{v} \rangle^2] = 1$, $\|\|h_{\mathbf{v}}\|\|_{\psi_2} \leq \tilde{\kappa}$ for every $h_{\mathbf{v}} \in \mathcal{H}$, and $\|\|h_{\mathbf{v}_1} - h_{\mathbf{v}_2}\|\|_{\psi_2} \leq \tilde{\kappa}\|\mathbf{v}_1 - \mathbf{v}_2\|_2$ for any $h_{\mathbf{v}_1}, h_{\mathbf{v}_2} \in \mathcal{H}$. Further, if we let $\beta = \frac{1}{2}$ and $mn \geq 4c_1 c_2 \tilde{\kappa}^4 w^2(\mathcal{A}_{\mathbf{\Lambda}}) \triangleq C_1 \tilde{\kappa}^4 w^2(\mathcal{A}_{\mathbf{\Lambda}})$, using (S.7), (S.8) and (S.9), we have

$$c_1 \tilde{\kappa} \gamma_2 \left(\mathcal{H}, \|\|\cdot\|\|_{\psi_2}\right) \leq c_1 \tilde{\kappa} \gamma_2 \left(\mathcal{A}_{\mathbf{\Lambda}}, \|\cdot\|_2\right) \leq c_1 c_4 \tilde{\kappa}^2 w(\mathcal{A}_{\mathbf{\Lambda}}) \leq \beta \sqrt{mn}$$

By Lemma C, with probability at least $1 - \exp(-c_2 \beta^2 mn/\tilde{\kappa}^4) \triangleq 1 - \exp(-C_2 mn/\tilde{\kappa}^4)$,

$$\sup_{h \in \mathcal{H}_j} \left| \frac{1}{mn} \sum_{i=1}^{n} \sum_{j=1}^{m} h^2(\tilde{\mathbf{x}}_i^j) - \mathbb{E}[h^2] \right| = \sup_{\mathbf{v} \in \mathcal{A}_{\mathbf{\Lambda}}} \left| \frac{1}{mn} \sum_{i=1}^{n} \mathbf{v}^T \tilde{\mathbf{X}}_i^T \tilde{\mathbf{X}}_i \mathbf{v} - 1 \right| \leq \frac{1}{2}$$

$$\implies \quad \frac{1}{n} \sum_{i=1}^{n} \mathbf{v}^T \tilde{\mathbf{X}}_i^T \tilde{\mathbf{X}}_i \mathbf{v} \geq \frac{m}{2}, \quad \forall \mathbf{v} \in \mathcal{A}_{\mathbf{\Gamma}_j}$$

$$\implies \quad \frac{1}{n} \sum_{i=1}^{n} \mathbf{v}^T \tilde{\mathbf{X}}_i^T \mathbf{\Xi}^{\frac{1}{2}} \mathbf{\Sigma}^{-1} \mathbf{\Xi}^{\frac{1}{2}} \tilde{\mathbf{X}}_i \mathbf{v} \geq \frac{m}{2} \cdot \lambda_{\min}\left(\mathbf{\Xi}^{\frac{1}{2}} \mathbf{\Sigma}^{-1} \mathbf{\Xi}^{\frac{1}{2}}\right), \quad \forall \mathbf{v} \in \mathcal{A}_{\mathbf{\Lambda}}$$

$$\implies \frac{1}{n} \sum_{i=1}^{n} \mathbf{v}^T \mathbf{\Lambda}^{-\frac{1}{2}} \mathbf{\Lambda}^{\frac{1}{2}} \tilde{\mathbf{X}}_i^T \mathbf{\Xi}^{\frac{1}{2}} \mathbf{\Sigma}^{-1} \mathbf{\Xi}^{\frac{1}{2}} \tilde{\mathbf{X}}_i \mathbf{\Lambda}^{\frac{1}{2}} \mathbf{\Lambda}^{-\frac{1}{2}} \mathbf{v} \geq \frac{m}{2} \cdot \lambda_{\min}\left(\mathbf{\Xi}^{\frac{1}{2}} \mathbf{\Sigma}^{-1} \mathbf{\Xi}^{\frac{1}{2}}\right) \mathbf{v}^T \mathbf{v}, \ \forall \mathbf{v} \in \mathcal{A}_{\mathbf{\Lambda}}$$

Now we replace $\boldsymbol{\Lambda}^{-\frac{1}{2}}\mathbf{v}$ by $\mathbf{w}$ and use the definition of $\mathcal{A}_{\boldsymbol{\Lambda}}$ to obtain

$$\frac{1}{n}\sum_{i=1}^{n}\mathbf{w}^{T}\boldsymbol{\Lambda}^{\frac{1}{2}}\tilde{\mathbf{X}}_{i}^{T}\boldsymbol{\Xi}^{\frac{1}{2}}\boldsymbol{\Sigma}^{-1}\boldsymbol{\Xi}^{\frac{1}{2}}\tilde{\mathbf{X}}_{i}\boldsymbol{\Lambda}^{\frac{1}{2}}\mathbf{w} \geq \frac{m}{2}\cdot\lambda_{\min}\left(\boldsymbol{\Xi}^{\frac{1}{2}}\boldsymbol{\Sigma}^{-1}\boldsymbol{\Xi}^{\frac{1}{2}}\right)\cdot\mathbf{w}^{T}\boldsymbol{\Lambda}\mathbf{w}, \quad \forall\,\mathbf{w}\in\mathrm{cone}(\mathcal{A})$$

$$\implies \quad \frac{1}{n}\sum_{i=1}^{n}\mathbf{w}^{T}\mathbf{X}_{i}^{T}\boldsymbol{\Sigma}^{-1}\mathbf{X}_{i}\mathbf{w} \geq \frac{m}{2}\cdot\lambda_{\min}\left(\boldsymbol{\Xi}^{\frac{1}{2}}\boldsymbol{\Sigma}^{-1}\boldsymbol{\Xi}^{\frac{1}{2}}\right)\cdot\lambda_{\min}\left(\boldsymbol{\Lambda}\right), \quad \forall\,\mathbf{w}\in\mathcal{A}$$

$$\implies \quad \mathbf{w}^{T}\hat{\boldsymbol{\Gamma}}\mathbf{w} \geq \frac{m}{2}\cdot\lambda_{\min}\left(\boldsymbol{\Xi}^{\frac{1}{2}}\boldsymbol{\Sigma}^{-1}\boldsymbol{\Xi}^{\frac{1}{2}}\right)\cdot\lambda_{\min}\left(\boldsymbol{\Lambda}\right), \quad \forall\,\mathbf{w}\in\mathcal{A}$$

Finally we need to bound the Gaussian width $w(\mathcal{A}_{\boldsymbol{\Lambda}})$. Note that the proof of Lemma 1 implies that $\|\boldsymbol{\Xi}^{\frac{1}{2}}\mathbf{u}\|_{2}^{2}\cdot\boldsymbol{\Lambda} = \mathbb{E}[\mathbf{X}^{T}\mathbf{u}\mathbf{u}^{T}\mathbf{X}] = \boldsymbol{\Gamma}_{\mathbf{u}}$ for any $\mathbf{u}\in\mathbb{S}^{p-1}$. Therefore it is not difficult to see that $\mathcal{A}_{\boldsymbol{\Lambda}} = \mathcal{A}_{\boldsymbol{\Gamma}_{\mathbf{u}}}$. Using Lemma 1 and 4, we have

$$w(\mathcal{A}_{\boldsymbol{\Lambda}}) = w(\mathcal{A}_{\boldsymbol{\Gamma}_{\mathbf{u}}}) \leq C\kappa_{0}\sqrt{\frac{\mu_{\max}}{\mu_{\min}}}\cdot(w(\mathcal{A})+3) = C\kappa_{0}\sqrt{\frac{\lambda_{\max}(\boldsymbol{\Xi})\lambda_{\max}(\boldsymbol{\Lambda})}{\lambda_{\min}(\boldsymbol{\Xi})\lambda_{\min}(\boldsymbol{\Lambda})}}\cdot(w(\mathcal{A})+3) ,$$

which completes the proof. ∎

## 2.8 Proof of Corollary 2

**Statement of Corollary 2:** *Suppose $\mathbf{y} = \mathbf{X}\boldsymbol{\theta}^{*} + \boldsymbol{\eta} \in \mathbb{R}^{m}$, where $\mathbf{X}$ is described in Lemma 6, and $\boldsymbol{\eta}\sim\mathcal{N}(\mathbf{0},\mathbf{I})$. With probability at least $1 - \exp\left(-\frac{m}{2}\right) - C_{2}\exp\left(-\frac{C_{1}^{2}w^{2}(\mathcal{B})}{4\rho^{2}}\right) - \exp\left(-C_{3}m/\tilde{\kappa}^{4}\right)$, $\hat{\boldsymbol{\theta}}_{sg}$ satisfies*

$$\left\|\hat{\boldsymbol{\theta}}_{sg}-\boldsymbol{\theta}^{*}\right\|_{2} \leq C\tilde{\kappa}\cdot\sqrt{\frac{\lambda_{\max}(\boldsymbol{\Xi})\lambda_{\max}(\boldsymbol{\Lambda})}{\lambda_{\min}^{2}(\boldsymbol{\Xi})\lambda_{\min}^{2}(\boldsymbol{\Lambda})}}\cdot\frac{\Psi(\boldsymbol{\theta}^{*})\cdot w(\mathcal{B})}{\sqrt{m}} , \tag{S.34}$$

*Proof:* Setting $n=1$ and $\boldsymbol{\Sigma} = \boldsymbol{\Sigma}_{*} = \mathbf{I}$ for Lemma 5, we have

$$\left\|\mathbf{X}^{T}\boldsymbol{\Sigma}^{-1}\boldsymbol{\eta}\right\|_{*} = \left\|\mathbf{X}^{T}\boldsymbol{\eta}\right\|_{*} \leq c\tilde{\kappa}\sqrt{m\cdot\mu_{\max}}\cdot w(\mathcal{B}) = c\tilde{\kappa}\sqrt{m\cdot\lambda_{\max}(\boldsymbol{\Xi})\lambda_{\max}(\boldsymbol{\Lambda})}\cdot w(\mathcal{B}) ,$$

with probability $1 - \exp\left(-\frac{m}{2}\right) - C_{2}\exp(-\frac{C_{1}^{2}w^{2}(\mathcal{B})}{4\rho^{2}})$. By Lemma 6, we have $\alpha = \frac{m\cdot\lambda_{\min}(\boldsymbol{\Xi})\lambda_{\min}(\boldsymbol{\Lambda})}{2}$, with probability at least $1 - \exp(-C_{3}m/\tilde{\kappa}^{4})$. Therefore, it follows from Lemma 2 that

$$\|\hat{\boldsymbol{\theta}}_{sg}-\boldsymbol{\theta}^{*}\|_{2} \leq 2\Psi(\boldsymbol{\theta}^{*})\cdot\frac{\gamma}{\alpha} \leq C\tilde{\kappa}\cdot\sqrt{\frac{\lambda_{\max}(\boldsymbol{\Xi})\lambda_{\max}(\boldsymbol{\Lambda})}{\lambda_{\min}^{2}(\boldsymbol{\Xi})\lambda_{\min}^{2}(\boldsymbol{\Lambda})}}\cdot\frac{\Psi(\boldsymbol{\theta}^{*})\cdot w(\mathcal{B})}{\sqrt{m}}$$

which completes the proof. ∎

# 3 Proofs for Section 3.2

## 3.1 Proof of Theorem 2

**Statement of Theorem 2:**
*If $n \geq C^{4}m\cdot\max\left\{4\left(\kappa_{0}+\kappa\sqrt{\frac{\mu_{\max}}{\lambda_{\min}(\boldsymbol{\Sigma}_{*})}}\|\boldsymbol{\theta}^{*}-\boldsymbol{\theta}\|_{2}\right)^{4}, \kappa^{4}\left(\frac{\lambda_{\max}(\boldsymbol{\Sigma}_{*})\mu_{\max}}{\lambda_{\min}(\boldsymbol{\Sigma}_{*})\mu_{\min}}\right)^{2}\right\}$ and $\mathbf{X}_{i}$ is sub-Gaussian, with probability at least $1 - 2\exp(-C_{1}m)$, $\hat{\boldsymbol{\Sigma}}$ given by (22) satisfies*

$$\lambda_{\max}\left(\boldsymbol{\Sigma}_{*}^{-\frac{1}{2}}\hat{\boldsymbol{\Sigma}}\boldsymbol{\Sigma}_{*}^{-\frac{1}{2}}\right) \leq 1 + C^{2}\kappa_{0}^{2}\sqrt{m/n} + \frac{2\mu_{\max}}{\lambda_{\min}(\boldsymbol{\Sigma}_{*})}\|\boldsymbol{\theta}^{*}-\boldsymbol{\theta}\|_{2}^{2} \tag{S.35}$$

$$\lambda_{\min}\left(\boldsymbol{\Sigma}_{*}^{-\frac{1}{2}}\hat{\boldsymbol{\Sigma}}\boldsymbol{\Sigma}_{*}^{-\frac{1}{2}}\right) \geq 1 - C^{2}\kappa_{0}^{2}\sqrt{m/n} \tag{S.36}$$

*Proof:* By introducing the true parameter $\boldsymbol{\theta}^{*}$, $\hat{\boldsymbol{\Sigma}}$ can be rewritten as

$$\hat{\boldsymbol{\Sigma}} = \frac{1}{n}\sum_{i=1}^{n}\left(\boldsymbol{\eta}_{i}+\mathbf{X}_{i}(\boldsymbol{\theta}^{*}-\boldsymbol{\theta})\right)\left(\boldsymbol{\eta}_{i}+\mathbf{X}_{i}(\boldsymbol{\theta}^{*}-\boldsymbol{\theta})\right)^{T}$$

And note that
$$\mathbf{\Sigma}_{\boldsymbol{\theta}} \triangleq \mathbb{E}[\hat{\mathbf{\Sigma}}] = \mathbf{\Sigma}_* + \mathbf{\Delta}_{\boldsymbol{\theta}}, \ \text{ where } \ \mathbf{\Delta}_{\boldsymbol{\theta}} = \mathbb{E}\left[\mathbf{X}(\boldsymbol{\theta}^* - \boldsymbol{\theta})(\boldsymbol{\theta}^* - \boldsymbol{\theta})^T \mathbf{X}^T\right].$$

The $\psi_2$-norm of $\mathbf{\Sigma}_*^{-\frac{1}{2}}\left(\boldsymbol{\eta} + \mathbf{X}(\boldsymbol{\theta}^* - \boldsymbol{\theta})\right)$ satisfies

$$\left|\!\left|\!\left|\mathbf{\Sigma}_*^{-\frac{1}{2}}\left(\boldsymbol{\eta} + \mathbf{X}(\boldsymbol{\theta}^* - \boldsymbol{\theta})\right)\right|\!\right|\!\right|_{\psi_2} \leq \left|\!\left|\!\left|\mathbf{\Sigma}_*^{-\frac{1}{2}}\boldsymbol{\eta}\right|\!\right|\!\right|_{\psi_2} + \left|\!\left|\!\left|\mathbf{\Sigma}_*^{-\frac{1}{2}}\mathbf{X}(\boldsymbol{\theta}^* - \boldsymbol{\theta})\right|\!\right|\!\right|_{\psi_2}$$

$$= \|\tilde{\boldsymbol{\eta}}\|_{\psi_2} + \sup_{\mathbf{u} \in \mathbb{S}^{m-1}} \left|\!\left|\!\left|(\boldsymbol{\theta}^* - \boldsymbol{\theta})^T \mathbf{\Gamma}_{*\mathbf{u}}^{\frac{1}{2}} \mathbf{\Gamma}_{*\mathbf{u}}^{-\frac{1}{2}} \mathbf{X}^T \mathbf{\Sigma}_*^{-\frac{1}{2}}\mathbf{u}\right|\!\right|\!\right|_{\psi_2}$$

$$\leq \kappa_0 + \sup_{\substack{\mathbf{v} \in \mathbb{S}^{p-1} \\ \mathbf{u} \in \mathbb{S}^{m-1}}} \left\|\mathbf{\Gamma}_{*\mathbf{u}}^{\frac{1}{2}}(\boldsymbol{\theta}^* - \boldsymbol{\theta})\right\|_2 \cdot \left|\!\left|\!\left|\mathbf{v}^T \mathbf{\Gamma}_{*\mathbf{u}}^{-\frac{1}{2}} \mathbf{X}^T \mathbf{\Sigma}_*^{-\frac{1}{2}}\mathbf{u}\right|\!\right|\!\right|_{\psi_2}$$

$$\leq \kappa_0 + \kappa \sup_{\mathbf{u} \in \mathbb{S}^{m-1}} \left\|\mathbf{\Gamma}_{*\mathbf{u}}^{\frac{1}{2}}\right\|_2 \|\boldsymbol{\theta}^* - \boldsymbol{\theta}\|_2$$

$$\leq \kappa_0 + \kappa \sqrt{\frac{\mu_{\max}}{\lambda_{\min}(\mathbf{\Sigma}_*)}} \|\boldsymbol{\theta}^* - \boldsymbol{\theta}\|_2$$

where $\mathbf{\Gamma}_{*\mathbf{u}} = \mathbb{E}[\mathbf{X}^T \mathbf{\Sigma}_*^{-\frac{1}{2}} \mathbf{u}\mathbf{u}^T \mathbf{\Sigma}_*^{-\frac{1}{2}} \mathbf{X}]$, and $\|\mathbf{\Gamma}_{*\mathbf{u}}\|_2^2 \leq \mu_{\max}\|\mathbf{\Sigma}_*^{-\frac{1}{2}}\mathbf{u}\|_2^2 \leq \frac{\mu_{\max}}{\lambda_{\min}(\mathbf{\Sigma}_*)}$ by the definition of sub-Gaussian $\mathbf{X}$. $\kappa_0$ is the $\psi_2$-norm of standard Gaussian random vector. By Theorem 5.39 and Remark 5.40 in [6], if $n \geq C_0^4 m \left(\kappa_0 + \kappa\sqrt{\frac{\mu_{\max}}{\lambda_{\min}(\mathbf{\Sigma}_*)}} \|\boldsymbol{\theta}^* - \boldsymbol{\theta}\|_2\right)^4$, with probability at least $1 - 2\exp(-C_1 m)$, we have

$$\left\|\mathbf{\Sigma}_*^{-\frac{1}{2}}\left(\hat{\mathbf{\Sigma}} - \mathbf{\Sigma}_{\boldsymbol{\theta}}\right)\mathbf{\Sigma}_*^{-\frac{1}{2}}\right\|_2 \leq C_0^2\left(\kappa_0 + \kappa\sqrt{\frac{\mu_{\max}}{\lambda_{\min}(\mathbf{\Sigma}_*)}} \|\boldsymbol{\theta}^* - \boldsymbol{\theta}\|_2\right)^2 \sqrt{\frac{m}{n}} \tag{S.37}$$

Hence we have

$$\lambda_{\max}\left(\mathbf{\Sigma}_*^{-\frac{1}{2}}\hat{\mathbf{\Sigma}}\mathbf{\Sigma}_*^{-\frac{1}{2}}\right) = \left\|\mathbf{\Sigma}_*^{-\frac{1}{2}}\hat{\mathbf{\Sigma}}\mathbf{\Sigma}_*^{-\frac{1}{2}}\right\|_2 \leq 1 + \left\|\mathbf{\Sigma}_*^{-\frac{1}{2}}\left(\hat{\mathbf{\Sigma}} - \mathbf{\Sigma}_{\boldsymbol{\theta}}\right)\mathbf{\Sigma}_*^{-\frac{1}{2}}\right\|_2 + \left\|\mathbf{\Sigma}_*^{-\frac{1}{2}}\mathbf{\Delta}_{\boldsymbol{\theta}}\mathbf{\Sigma}_*^{-\frac{1}{2}}\right\|_2$$

$$\leq 1 + C_0^2\left(\kappa_0 + \kappa\sqrt{\frac{\mu_{\max}}{\lambda_{\min}(\mathbf{\Sigma}_*)}} \|\boldsymbol{\theta}^* - \boldsymbol{\theta}\|_2\right)^2 \sqrt{\frac{m}{n}} + \frac{\mu_{\max}}{\lambda_{\min}(\mathbf{\Sigma}_*)} \|\boldsymbol{\theta}^* - \boldsymbol{\theta}\|_2^2$$

$$\overset{(a)}{\leq} 1 + 2C_0^2\kappa_0^2\sqrt{\frac{m}{n}} + \frac{2C_0^2\kappa^2\mu_{\max}}{\lambda_{\min}(\mathbf{\Sigma}_*)} \|\boldsymbol{\theta}^* - \boldsymbol{\theta}\|_2^2 \sqrt{\frac{m}{n}} + \frac{\mu_{\max}}{\lambda_{\min}(\mathbf{\Sigma}_*)} \|\boldsymbol{\theta}^* - \boldsymbol{\theta}\|_2^2$$

$$\leq 1 + 2C_0^2\kappa_0^2\sqrt{\frac{m}{n}} + \left(\frac{\mu_{\min}}{\lambda_{\max}(\mathbf{\Sigma}_*)} + \frac{\mu_{\max}}{\lambda_{\min}(\mathbf{\Sigma}_*)}\right) \|\boldsymbol{\theta}^* - \boldsymbol{\theta}\|_2^2$$

$$\leq 1 + C^2\kappa_0^2\sqrt{\frac{m}{n}} + \frac{2\mu_{\max}}{\lambda_{\min}(\mathbf{\Sigma}_*)} \|\boldsymbol{\theta}^* - \boldsymbol{\theta}\|_2^2 \tag{S.38}$$

$$\lambda_{\min}\left(\mathbf{\Sigma}_*^{-\frac{1}{2}}\hat{\mathbf{\Sigma}}\mathbf{\Sigma}_*^{-\frac{1}{2}}\right) \geq 1 + \lambda_{\min}\left(\mathbf{\Sigma}_*^{-\frac{1}{2}}\left(\hat{\mathbf{\Sigma}} - \mathbf{\Sigma}_{\boldsymbol{\theta}}\right)\mathbf{\Sigma}_*^{-\frac{1}{2}}\right) + \lambda_{\min}\left(\mathbf{\Sigma}_*^{-\frac{1}{2}}\mathbf{\Delta}_{\boldsymbol{\theta}}\mathbf{\Sigma}_*^{-\frac{1}{2}}\right)$$

$$\geq 1 - \left\|\mathbf{\Sigma}_*^{-\frac{1}{2}}\left(\hat{\mathbf{\Sigma}} - \mathbf{\Sigma}_{\boldsymbol{\theta}}\right)\mathbf{\Sigma}_*^{-\frac{1}{2}}\right\|_2 + \frac{\mu_{\min}}{\lambda_{\max}(\mathbf{\Sigma}_*)} \|\boldsymbol{\theta}^* - \boldsymbol{\theta}\|_2^2$$

$$\geq 1 - C_0^2\left(\kappa_0 + \kappa\sqrt{\frac{\mu_{\max}}{\lambda_{\min}(\mathbf{\Sigma}_*)}} \|\boldsymbol{\theta}^* - \boldsymbol{\theta}\|_2\right)^2 \sqrt{\frac{m}{n}} + \frac{\mu_{\min}}{\lambda_{\max}(\mathbf{\Sigma}_*)} \|\boldsymbol{\theta}^* - \boldsymbol{\theta}\|_2^2$$

$$\overset{(b)}{\geq} 1 - 2C_0^2\kappa_0^2\sqrt{\frac{m}{n}} - \frac{2C_0^2\kappa^2\mu_{\max}}{\lambda_{\min}(\mathbf{\Sigma}_*)} \|\boldsymbol{\theta}^* - \boldsymbol{\theta}\|_2^2 \sqrt{\frac{m}{n}} + \frac{\mu_{\min}}{\lambda_{\max}(\mathbf{\Sigma}_*)} \|\boldsymbol{\theta}^* - \boldsymbol{\theta}\|_2^2$$

$$\geq 1 - C^2\kappa_0^2\sqrt{\frac{m}{n}} \tag{S.39}$$

where $C^2 = 2C_0^2$, and in both (a) and (b), we use the assumption $n \geq C^4 m \kappa^4 \left(\frac{\lambda_{\max}(\mathbf{\Sigma}_*)\mu_{\max}}{\lambda_{\min}(\mathbf{\Sigma}_*)\mu_{\min}}\right)^2 = 4C_0^4 m \kappa^4 \left(\frac{\lambda_{\max}(\mathbf{\Sigma}_*)\mu_{\max}}{\lambda_{\min}(\mathbf{\Sigma}_*)\mu_{\min}}\right)^2$. This completes the proof. ∎

# 4 Proofs for Section 3.3

## 4.1 Proof of Lemma 7

**Statement of Lemma 7:** *If $\hat{\Sigma}$ is given as (22) and the condition in Theorem 2 holds, then the inequality below holds with probability at least $1 - 2\exp(-C_1 m)$,*

$$\xi\left(\hat{\Sigma}\right) \leq \xi\left(\Sigma_*\right) \cdot \left(1 + 2C\kappa_0 \left(\frac{m}{n}\right)^{\frac{1}{4}} + 2\sqrt{\frac{\mu_{\max}}{\lambda_{\min}\left(\Sigma_*\right)}} \left\|\theta^* - \theta\right\|_2\right) \tag{S.40}$$

*Proof:* Based on the definition of $\xi(\cdot)$, we have

$$\xi\left(\hat{\Sigma}\right) = \frac{\sqrt{\text{Tr}\left(\hat{\Sigma}^{-1}\Sigma_*\hat{\Sigma}^{-1}\right)}}{\text{Tr}\left(\hat{\Sigma}^{-1}\right)} = \frac{1}{\sqrt{\text{Tr}\left(\Sigma_*^{-1}\right)}} \cdot \sqrt{\frac{\text{Tr}\left(\Sigma_*^{-1}\right) \cdot \text{Tr}\left(\hat{\Sigma}^{-1}\Sigma_*\hat{\Sigma}^{-1}\right)}{\text{Tr}^2\left(\hat{\Sigma}^{-1}\right)}}$$

$$= \xi\left(\Sigma_*\right) \cdot \sqrt{\frac{\text{Tr}\left(\hat{\Sigma}^{\frac{1}{2}}\Sigma_*^{-1}\hat{\Sigma}^{\frac{1}{2}}\hat{\Sigma}^{-1}\right) \cdot \text{Tr}\left(\hat{\Sigma}^{-\frac{1}{2}}\Sigma_*\hat{\Sigma}^{-\frac{1}{2}}\hat{\Sigma}^{-1}\right)}{\text{Tr}^2\left(\hat{\Sigma}^{-1}\right)}}$$

$$\leq \xi\left(\Sigma_*\right) \cdot \sqrt{\frac{\lambda_{\max}\left(\hat{\Sigma}^{\frac{1}{2}}\Sigma_*^{-1}\hat{\Sigma}^{\frac{1}{2}}\right)\text{Tr}\left(\hat{\Sigma}^{-1}\right) \cdot \lambda_{\max}\left(\hat{\Sigma}^{-\frac{1}{2}}\Sigma_*\hat{\Sigma}^{-\frac{1}{2}}\right)\text{Tr}\left(\hat{\Sigma}^{-1}\right)}{\text{Tr}^2\left(\hat{\Sigma}^{-1}\right)}}$$

$$= \xi\left(\Sigma_*\right) \cdot \sqrt{\lambda_{\max}\left(\hat{\Sigma}^{\frac{1}{2}}\Sigma_*^{-1}\hat{\Sigma}^{\frac{1}{2}}\right)\lambda_{\max}\left(\hat{\Sigma}^{-\frac{1}{2}}\Sigma_*\hat{\Sigma}^{-\frac{1}{2}}\right)} = \xi\left(\Sigma_*\right) \cdot \sqrt{\frac{\lambda_{\max}\left(\Sigma_*^{-\frac{1}{2}}\hat{\Sigma}\Sigma_*^{-\frac{1}{2}}\right)}{\lambda_{\min}\left(\Sigma_*^{-\frac{1}{2}}\hat{\Sigma}\Sigma_*^{-\frac{1}{2}}\right)}} \tag{S.41}$$

where the inequality follows from von Neumann's trace inequality. Now we can bound $\xi(\hat{\Sigma})$ by invoking Theorem 2,

$$\xi\left(\hat{\Sigma}\right) \leq \xi\left(\Sigma_*\right) \cdot \sqrt{\frac{1 + C^2\kappa_0^2\sqrt{\frac{m}{n}} + \frac{2\mu_{\max}}{\lambda_{\min}(\Sigma_*)}\left\|\theta^* - \theta\right\|_2^2}{1 - C^2\kappa_0^2\sqrt{\frac{m}{n}}}}$$

$$= \xi\left(\Sigma_*\right) \cdot \sqrt{1 + \frac{2C^2\kappa_0^2\sqrt{\frac{m}{n}} + \frac{2\mu_{\max}}{\lambda_{\min}(\Sigma_*)}\left\|\theta^* - \theta\right\|_2^2}{1 - C^2\kappa_0^2\sqrt{\frac{m}{n}}}}$$

$$\leq \xi\left(\Sigma_*\right) \cdot \left(1 + \frac{\sqrt{2}C\kappa_0\left(\frac{m}{n}\right)^{\frac{1}{4}} + \sqrt{\frac{2\mu_{\max}}{\lambda_{\min}(\Sigma_*)}}\left\|\theta^* - \theta\right\|_2}{\sqrt{1 - C^2\kappa_0^2\sqrt{\frac{m}{n}}}}\right) \tag{S.42}$$

$$\leq \xi\left(\Sigma_*\right) \cdot \left(1 + 2C\kappa_0\left(\frac{m}{n}\right)^{\frac{1}{4}} + 2\sqrt{\frac{\mu_{\max}}{\lambda_{\min}\left(\Sigma_*\right)}}\left\|\theta^* - \theta\right\|_2\right)$$

where the last inequality follows from $n \geq 4C^4 m \cdot \left(\kappa_0 + \kappa\sqrt{\frac{\mu_{\max}}{\lambda_{\min}(\Sigma_*)}}\left\|\theta^* - \theta\right\|_2\right)^4 \geq 4C^4 m\kappa_0^4$. ∎

## 4.2 Proof of Theorem 3

**Statement of Theorem 3:**

*Let $e_{orc} = C_1\kappa\sqrt{\frac{\mu_{\max}}{\mu_{\min}^2}}\frac{\xi(\Sigma_*)\cdot\Psi(\theta^*)w(\mathcal{B})}{\sqrt{n}}$ and $e_{\min} = e_{orc} \cdot \frac{1 + 2C\kappa_0\left(\frac{m}{n}\right)^{\frac{1}{4}}}{1 - 2e_{orc}\sqrt{\frac{\mu_{\max}}{\lambda_{\min}(\Sigma_*)}}}$. If $n \geq C^4 m \cdot$*

$$\max\left\{4\left(\kappa_0 + \frac{C_1}{C^2}\sqrt{\frac{\lambda_{\min}(\Sigma_*)}{\lambda_{\max}^2(\Sigma_*)}}\frac{\Psi(\theta^*)w(\mathcal{B})}{m}\right)^4, \ \kappa^4\left(\frac{\lambda_{\max}(\Sigma_*)\mu_{\max}}{\lambda_{\min}(\Sigma_*)\mu_{\min}}\right)^2, \ \left(\frac{2C_1\kappa\mu_{\max}}{C^2\mu_{\min}} \cdot \frac{\xi(\Sigma_*)\Psi(\theta^*)w(\mathcal{B})}{\sqrt{m}\cdot\lambda_{\min}(\Sigma_*)}\right)^2\right\}$$

*and also satisfies the condition in Theorem 1, with high probability, the iterate $\hat{\boldsymbol{\theta}}_T$ returned by Algorithm 1 satisfies*

$$\left\|\hat{\boldsymbol{\theta}}_T - \boldsymbol{\theta}^*\right\|_2 \leq e_{min} + \left(2e_{orc}\sqrt{\frac{\mu_{\max}}{\lambda_{\min}\left(\boldsymbol{\Sigma}_*\right)}}\right)^{T-1} \cdot \left(\left\|\hat{\boldsymbol{\theta}}_1 - \boldsymbol{\theta}^*\right\|_2 - e_{min}\right) \qquad \text{(S.43)}$$

*Proof:* Since $n \geq C^4 m \kappa^4 \left(\frac{\lambda_{\max}(\boldsymbol{\Sigma}_*)\mu_{\max}}{\lambda_{\min}(\boldsymbol{\Sigma}_*)\mu_{\min}}\right)^2$ and $\hat{\boldsymbol{\Sigma}}_0$ is initialized as $\hat{\boldsymbol{\Sigma}}_0 = \mathbf{I}_{m \times m}$, by applying Theorem 1 to $\hat{\boldsymbol{\theta}}_1$, we have

$$\begin{aligned}
\left\|\hat{\boldsymbol{\theta}}_1 - \boldsymbol{\theta}^*\right\|_2 &\leq C_1\kappa\sqrt{\frac{\mu_{\max}}{\mu_{\min}^2}}\cdot\xi\left(\hat{\boldsymbol{\Sigma}}_0\right)\cdot\frac{\Psi(\boldsymbol{\theta}^*)\cdot w(\mathcal{B})}{\sqrt{m}} = C_1\kappa\sqrt{\frac{\mu_{\max}}{\mu_{\min}^2}}\cdot\frac{\Psi(\boldsymbol{\theta}^*)\cdot w(\mathcal{B})}{\sqrt{mn}}\\
&\leq C_1\kappa\sqrt{\frac{\mu_{\max}}{\mu_{\min}^2}}\cdot\frac{\Psi(\boldsymbol{\theta}^*)\cdot w(\mathcal{B})}{\sqrt{m}}\cdot\frac{\lambda_{\min}\left(\boldsymbol{\Sigma}_*\right)\mu_{\min}}{C^2\sqrt{m}\cdot\kappa^2\lambda_{\max}\left(\boldsymbol{\Sigma}_*\right)\mu_{\max}}\\
&= \frac{C_1}{C^2}\cdot\frac{\lambda_{\min}\left(\boldsymbol{\Sigma}_*\right)}{\kappa\lambda_{\max}\left(\boldsymbol{\Sigma}_*\right)\sqrt{\mu_{\max}}}\cdot\frac{\Psi(\boldsymbol{\theta}^*)\cdot w(\mathcal{B})}{m}
\end{aligned}$$

It follows that

$$n \geq C^4 m \cdot 4\left(\kappa_0 + \frac{C_1}{C^2}\sqrt{\frac{\lambda_{\min}\left(\boldsymbol{\Sigma}_*\right)}{\lambda_{\max}^2\left(\boldsymbol{\Sigma}_*\right)}}\frac{\Psi(\boldsymbol{\theta}^*)w(\mathcal{B})}{m}\right)^4 \implies$$

$$n \geq C^4 m \cdot 4\left(\kappa_0 + \kappa\sqrt{\frac{\mu_{\max}}{\lambda_{\min}\left(\boldsymbol{\Sigma}_*\right)}}\left\|\boldsymbol{\theta}^* - \hat{\boldsymbol{\theta}}_1\right\|_2\right)^4$$

By applying Lemma 7 and Theorem 1 to the second iteration,

$$\left\|\hat{\boldsymbol{\theta}}_2 - \boldsymbol{\theta}^*\right\|_2 \leq e_{orc}\cdot\left(1 + 2C\kappa_0\left(\frac{m}{n}\right)^{\frac{1}{4}} + 2\sqrt{\frac{\mu_{\max}}{\lambda_{\min}\left(\boldsymbol{\Sigma}_*\right)}}\left\|\hat{\boldsymbol{\theta}}_1 - \boldsymbol{\theta}^*\right\|_2\right) \implies$$

$$\left\|\hat{\boldsymbol{\theta}}_2 - \boldsymbol{\theta}^*\right\|_2 - e_{min} \leq 2e_{orc}\sqrt{\frac{\mu_{\max}}{\lambda_{\min}\left(\boldsymbol{\Sigma}_*\right)}}\cdot\left(\left\|\hat{\boldsymbol{\theta}}_1 - \boldsymbol{\theta}^*\right\|_2 - e_{min}\right).$$

Since $n \geq C^4 m \cdot \left(\frac{2C_1\kappa}{C^2}\cdot\frac{\mu_{\max}}{\mu_{\min}}\cdot\frac{\xi(\boldsymbol{\Sigma}_*)\Psi(\boldsymbol{\theta}^*)w(\mathcal{B})}{\sqrt{m}\cdot\lambda_{\min}(\boldsymbol{\Sigma}_*)}\right)^2$, we have $2e_{orc}\sqrt{\frac{\mu_{\max}}{\lambda_{\min}(\boldsymbol{\Sigma}_*)}} \leq 1$, which indicates that $\left\|\hat{\boldsymbol{\theta}}_2 - \boldsymbol{\theta}^*\right\|_2 \leq \left\|\hat{\boldsymbol{\theta}}_1 - \boldsymbol{\theta}^*\right\|_2$. Therefore the condition in Lemma 7 on sample size $n$ also holds for $\hat{\boldsymbol{\theta}}_2$ and so on. By repeatedly applying Lemma 7 and Theorem 1, we have the following inequality for every $t > 0$,

$$\left\|\hat{\boldsymbol{\theta}}_{t+1} - \boldsymbol{\theta}^*\right\|_2 - e_{min} \leq 2e_{orc}\sqrt{\frac{\mu_{\max}}{\lambda_{\min}\left(\boldsymbol{\Sigma}_*\right)}}\cdot\left(\left\|\hat{\boldsymbol{\theta}}_t - \boldsymbol{\theta}^*\right\|_2 - e_{min}\right) \qquad \text{(S.44)}$$

By combining (S.44) for every $t$, we obtain

$$\left\|\hat{\boldsymbol{\theta}}_T - \boldsymbol{\theta}^*\right\|_2 - e_{min} \leq \left(2e_{orc}\sqrt{\frac{\mu_{\max}}{\lambda_{\min}\left(\boldsymbol{\Sigma}_*\right)}}\right)^{T-1}\cdot\left(\left\|\hat{\boldsymbol{\theta}}_1 - \boldsymbol{\theta}^*\right\|_2 - e_{min}\right)$$

which completes the proof. ∎