[Reviews · NeurIPS 2017]

Reviewer 1



Bounds on the estimation error in a multi-response structured Gaussian linear model is considered, where the parameter to be estimated is shared by all m responses and where the noise covariance may be different from identity. The authors propose an alternating procedure for estimation of the coefficient vector that essentially extends the GDS (generalized Dantzig selector). The new procedure accommodates general "low-complexity" structures on the parameter and non-identity covariance of the noise term. The paper is outside my field of expertise, so I apologize in advance for any mistakes due to my misunderstanding. The paper is very technical, and without the appropriate background, I was not able to really appreciate the significance of the contribution. That said, the theme of the results is quite clear, especially with the effort of the authors to supplement the results with intuitive explanations. 1. The paper is very technical, maybe too technical, and it is hard for a non-expert like myself to make sense of the conditions/assumptions made in the different theorems (despite the authors' evident effort to help with making sense of these conditions). 2. Again, I was not able to appreciate if the different assumptions and results are sensible or whether they are mostly ad-hoc or artifacts of the proofs, but overall the contribution over GDS and existing methods does not appear to be sufficiently significant for acceptance to NIPS. 3. The manuscript is not free of grammatical errors or bad wording, for example: -l.168: "at the same order" -caption of Fig 2: "The plots for oracle...implies..." -caption of Fig 2: "keep unchanged" (better: "remain unchanged") ## comments after reading author rebuttal: The authors have provided a clear non-technical summary of the paper, which is much appreciated. I would like to apologize for not being able to give a fair evaluation when originally reviewing the paper (this is because I did not have the technical background to appreciate the results).

Reviewer 2



The paper studies the problem of estimation in high dimensional multiple response regression. The author proposes a new estimation procedure, and give a non-asymptotic finite sample error bound, which is the first for this type of problem, except for related and more restricted non-asymptotic guarantees shown in [21]. Overall, the paper is clearly written and seems to provide new interesting results for this problem. The problem seems very related to multiple regression problem in statistics, where some structure is assumed to hold for the beta coefficients for different responses. It would be good to refer to the relevant statistics literature. The upper-bound on estimation error in Theorem 3 is complicated and involves many terms. It is not clear if all of them are necessary/tight, how does each one affect the actual estimation error etc. The authors also show simulation results, showing favourable accuracy close to an oracle procedure. It is however not explained what is the oracle estimator, defined in eq. (27), assuming, and why is it impossible to use it in practice. Minor: Line 248: Throughout out -> Throughout

Reviewer 3



The authors propose an alternating estimation (AltEst) procedure to estimate the model parameters of the problem of learning high-dimensional multiresponse linear models, based on the generalized Dantzig selector (GDS). Though it is a straightforward way to "solve" a non-convex problem, the authors also show its statistical guarantee and verify the results with experiments. I would say that showing the alternating estimation procedure is statistically sound might be more interesting that the problem itself, and may help us understanding some other similar problems.